# Control of the structural landscape and neuronal proteotoxicity of mutant Huntingtin by domains flanking the polyQ tract

Koning Shen[1,2], Barbara Calamini[3†], Jonathan A Fauerbach[1,2‡], Boxue Ma[4], Sarah H Shahmoradian[4§], Ivana L Serrano Lachapel[1,2], Wah Chiu[4], Donald C Lo[3], Judith Frydman[1,2*]

[1]Department of Biology, Stanford University, Stanford, United States; [2]Department of Genetics, Stanford University, Stanford, United States; [3]Center for Drug Discovery, Department of Neurobiology, Duke University Medical Center, Durham, United States; [4]Verna and Marrs McLean Department of Biochemistry and Molecular Biology, Baylor College of Medicine, Houston, United States

*For correspondence: jfrydman@
stanford.edu

Present address: †Open Innovation Access Platform, Sanofi Recherche and Développement, Strasbourg, France; ‡Miltenyi Biotec, Cologne, Germany; §Laboratory of Biomolecular Research, Paul Scherrer Institute, Villigen, Switzerland

Competing interests: The authors declare that no competing interests exist.

**Abstract** Many neurodegenerative diseases are linked to amyloid aggregation. In Huntington's disease (HD), neurotoxicity correlates with an increased aggregation propensity of a polyglutamine (polyQ) expansion in exon 1 of mutant huntingtin protein (mHtt). Here we establish how the domains flanking the polyQ tract shape the mHtt conformational landscape in vitro and in neurons. In vitro, the flanking domains have opposing effects on the conformation and stabilities of oligomers and amyloid fibrils. The N-terminal N17 promotes amyloid fibril formation, while the C-terminal Proline Rich Domain destabilizes fibrils and enhances oligomer formation. However, in neurons both domains act synergistically to engage protective chaperone and degradation pathways promoting mHtt proteostasis. Surprisingly, when proteotoxicity was assessed in rat corticostriatal brain slices, either flanking region alone sufficed to generate a neurotoxic conformation, while the polyQ tract alone exhibited minimal toxicity. Linking mHtt structural properties to its neuronal proteostasis should inform new strategies for neuroprotection in polyQ-expansion diseases.

## Introduction

Huntington's disease (HD) is an inherited neurodegenerative disease characterized by movement disorders, behavioral abnormalities, and brain atrophy (*Orr and Zoghbi, 2007*; *Ross et al., 2014*; *Vonsattel and DiFiglia, 1998*). HD arises from mutations in Huntingtin (Htt) that expand a polyglutamine (polyQ)-encoding CAG repeat in exon 1 above a threshold length of 35 Qs (*MacDonald, 1993*). The length of the polyQ tract correlates with both disease severity and propensity to form amyloid aggregates (*Scherzinger et al., 1999*). A link between neuronal toxicity and amyloid aggregation is further supported by post-mortem analyses of HD brains, which contain amyloid aggregates formed primarily by N-terminal exon 1 truncations of Htt (*Difiglia, 1997*; *Landles et al., 2010*; *Mangiarini et al., 1996*; *Ross and Poirier, 2004*). Such N-terminal fragments may arise from aberrant splicing at the Htt exon 1 junction or from caspase cleavage (*Sathasivam et al., 2013*; *Wellington et al., 2002*). Importantly, mutant Htt exon 1 carrying a polyQ expansion (herein 'mHtt-Ex1') suffices to cause HD-like disease in animal models (*Goldberg et al., 1996*; *Sathasivam et al.,*

**eLife digest** Huntington's disease is a neurodegenerative disorder in which misshapen proteins accumulate in the brain and kill neurons. The misshapen proteins form as a result of specific mutations in the gene that encodes a protein called huntingtin. These mutations result in a region of the protein called the polyQ tract being longer than normal. Other regions of huntingtin that are near to the polyQ tract can dramatically change the behavior of the mutant protein. Shen et al. investigated how these regions control the shape of mutant huntingtin and how this affects the toxicity of the mutant protein in neurons.

The experiments found that the two regions on either side of the polyQ tract dramatically change the shape of mutant huntingtin proteins. In the absence of these flanking regions, the extended polyQ region is not very toxic, demonstrating that the flanking sequences play important roles in generating the toxic protein shapes. These flanking regions help mutant huntingtin to form a particular shape that was strongly linked with the death of neurons in rat brain slices. The flanking regions also change the way that the cellular machinery in neurons recognizes mutated huntingtin proteins and acts to prevent them from causing harm.

Misshapen forms of other proteins are responsible for causing other neurodegenerative diseases, including Alzheimer's and Parkinson's diseases. Therefore, the findings of Shen et al. may help researchers to develop new drugs for these conditions, as well as for Huntingdon's disease.

*2013*; *Wellington et al., 2002*) and is thus widely used as a relevant model for HD biology and pathology.

Despite the link between aggregation and neurodegeneration, the underlying neurotoxic species in HD and other amyloid-linked neurodegenerative diseases such as Alzheimer's disease, Parkinson's disease, and amyotrophic lateral sclerosis remains elusive. One model proposes that amyloid aggregates are toxic because they sequester and deplete essential cellular proteins such as transcription factors or molecular chaperones (*Kirstein-Miles et al., 2013*; *Olzscha et al., 2011*). However, a number of findings question the causal relationship between mHtt amyloid aggregates and toxicity. For instance, the medium spiny striatal neurons more vulnerable to HD toxicity show few to no aggregates in HD patient brains. In contrast, less affected neuronal cell-types contain many aggregates (*Kuemmerle et al., 1999*). In addition, while transgenic mHtt-Ex1 is neurotoxic in mice, a longer N-terminal mHtt fragment encompassing exons 1 and 2 forms many aggregates in transgenic mice but exhibits no neuronal dysfunction (*Slow et al., 2005*). Finally, longitudinal survival studies of primary neurons expressing fluorescently-tagged mHtt-Ex1 demonstrated that the formation of an amyloid inclusion correlated with neuronal survival (*Arrasate et al., 2004*). Since fluorescence and EM imaging studies have suggested that mHtt can be sequestered in different types of cellular inclusions (*Caron et al., 2014*; *Lu et al., 2013*; *Nekooki-Machida et al., 2009*; *Sahl et al., 2016*) one possible explanation for these observations is that only some inclusions are toxic, while others are protective.

An alternative hypothesis proposes that toxicity resides in soluble oligomeric mHtt conformations (*Behrends et al., 2006*; *Campioni et al., 2010*; *Kim et al., 2016*; *Miller et al., 2011*; *Sun et al., 2015*). Biophysical studies indicate that amyloidogenic proteins, including mHtt, not only form amyloid fibrils but also populate an array of ill-defined soluble, oligomeric conformations. Studies with conformationally-sensitive antibodies indicate that these states are conformationally highly heterogeneous (*Duim et al., 2014*; *Kayed and Glabe, 2006*; *Nucifora et al., 2012*; *Sontag et al., 2012*). The transient and structurally diverse nature of these soluble species has hindered their characterization; accordingly, the nature and determinants involved in the formation of these oligomeric mHtt species remain elusive. It remains imperative to clarify the mHtt conformational landscape and link the various species – aggregates, oligomers, or even aberrant monomers – to proteotoxicity. We also must understand how various mHtt species engage the cellular protein homeostasis (or 'proteostasis') pathways that clear aberrant and aggregation-prone conformations (*Martinez-Vicente et al., 2010*; *Ravikumar et al., 2004*; *Rubinsztein et al., 2012*; *Rui et al., 2015*; *Tam et al., 2006*; *Tsvetkov et al., 2013*). mHtt interactions with the chaperones and degradation pathways that

handle misfolded proteins in the cell (*Balch et al., 2008*; *Hartl et al., 2011*) are likely key determinants of the cellular balance of toxic and non-toxic conformers.

Recent studies indicate that two domains flanking the expanded polyQ tract greatly influence mHtt aggregation and biology. N17, the 17 amino acid N-terminal flanking domain, has been shown to enhance mHtt aggregation (*Tam et al., 2009*; *Thakur et al., 2009*). N17 also mediates mHtt interaction with chaperones, hosts many post-translational modifications regulating mHtt toxicity (*Gu et al., 2009*; *Steffan, 2004*; *Thompson et al., 2009*), and harbors a functional nuclear export sequence (*Maiuri et al., 2013*; *Rockabrand et al., 2007*; *Zheng et al., 2013*). The C-terminal proline-rich flanking domain (herein 'PRD') also binds cellular factors and influences mHtt toxicity in yeast (*Duennwald et al., 2006*; *Gao et al., 2014*). The PRD has also been shown to slow aggregation (*Bhattacharyya et al., 2006*; *Tam et al., 2009*). Importantly, an integrated understanding of the interplay between the biophysical and cellular modulation of mHtt by these flanking regions should provide insights into the nature of proteotoxicity.

Here we combine biophysical and cell biological approaches to define how the domains flanking the polyQ tract modulate the ensemble of mHtt conformations in vitro and in vivo. Importantly, we link the mHtt conformational ensemble to mHtt proteostasis and toxicity in cultured neurons and brain slices. Biophysical and structural analyses demonstrate that N17 and PRD have opposing effects on the energetic barriers dictating the formation of aggregates and oligomers by mHtt both in vivo and in vitro. This interplay between N17 and PRD determines the formation of toxic mHtt conformations and their interaction with cellular proteostasis pathways. One corollary of our data is that neuronal mHtt toxicity cannot be explained by a simple model whereby amyloid fibrils or oligomers are toxic, but rather one that points to specific toxic conformational sub-populations. Our work linking the mHtt conformational landscape with neuronal proteostasis and toxicity informs rational avenues to leverage the roles of the polyQ flanking regions for HD therapeutics.

## Results

### Opposing effects of N17 and PRD flanking domains on polyQ amyloid propensity and aggregation

To evaluate the impact of N17 and PRD on the expanded polyQ tract, we created a set of mHtt-Ex1 deletion variants containing a pathogenic-length polyQ tract (Q51) and lacking the N17 (ΔN), PRD (ΔP), or both N17 and PRD domains (ΔNΔP). These variants were compared to an otherwise identical mHtt-Ex1 with the same pathogenic polyQ tract but containing both flanking domains (Ex1) (*Figure 1A*). The mHtt-Ex1 variants were recombinantly expressed and purified as soluble N-terminal GST fusion proteins (*Figure 1—figure supplement 1*); aggregation is initiated by cleavage of the GST moiety as previously described (*Scherzinger et al., 1999*; *Tam et al., 2006*).

We initially characterized how these flanking regions contribute to the aggregation kinetics of purified mHtt through two complementary approaches. The filter trap assay detects large, SDS-insoluble aggregates through filtration through a 20 μm cellulose acetate membrane, (*Wanker et al., 1999*). Formation of β-sheet rich amyloid structures was detected using the fluorescent dye ThioflavinT (*LeVine, 1999*). Consistent with previous reports, deletion of N17 reduces the rate and yield of amyloid formation and aggregation, while deletion of the PRD domain enhances formation of amyloids and aggregates (*Figure 1B–C*, *Figure 1—figure supplement 1B–C*) (*Crick et al., 2013*; *Tam et al., 2009*; *Thakur et al., 2009*). ΔNΔP exhibited a complex, combined behavior of both N17 and PRD deletions. Similar to ΔP, the ΔNΔP mutant displayed enhanced aggregation rates measured in the filter trap assay (*Figure 1B*). On the other hand, similar to ΔN, ΔNΔP had very slow kinetics of amyloid formation as measured by the ThioflavinT assay (*Figure 1C*). Indeed, fitting the normalized ThioflavinT amyloid formation kinetics to the Finke-Watzky kinetic model (*Alvarez et al., 2013*; *Morris et al., 2009*) showed that the elongation rate $v$ for amyloid formation and half-time to saturation of amyloid formation $t_{1/2}$ for mHtt variants lacking N17 (ΔN and ΔNΔP) were much slower than those for mHtt variants containing N17 (Ex1 and ΔP) (*Figure 1D*, *Figure 1—figure supplement 1D*). We conclude that the presence of a PRD disfavors formation of large, SDS-insoluble aggregates while N17 exerts a dominant effect to promote the ThioflavinT-reactive, amyloid conformation.

To relate these biophysical observations to mHtt-Ex1 behavior in a neuronal cellular environment, the equivalent mHtt-Ex1 variants were fused C-terminally to GFP and expressed in striatal neuron-

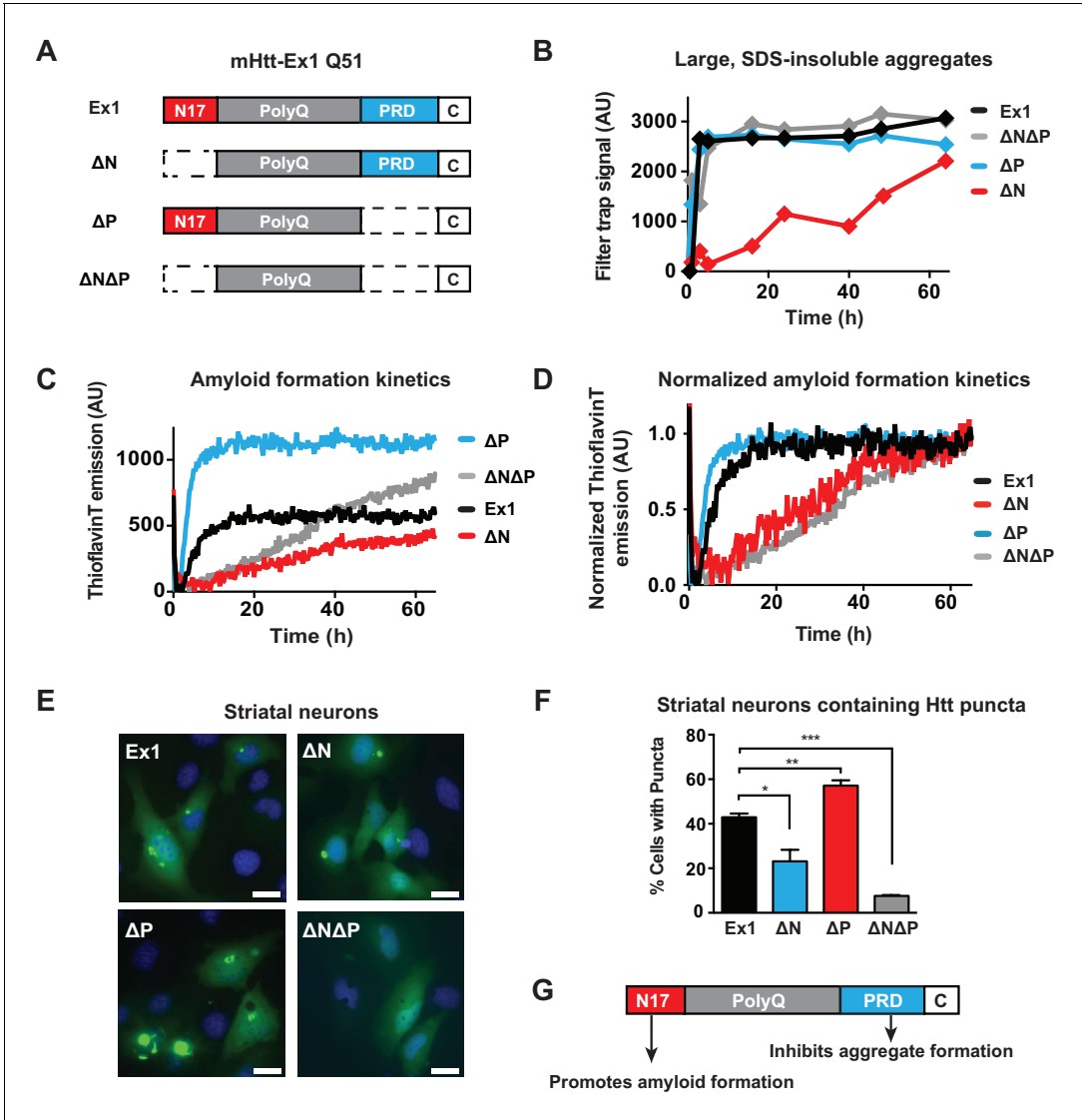

**Figure 1.** Flanking regions impact pathogenic mHtt aggregation propensity in vitro and in striatal neurons. (**A**) Schematic representation of polyQ expanded mutant Htt-Ex1 variants (mHtt-Ex1) used in this study. mHtt-Ex1 contains N17 (red), an expanded polyQ tract (grey), the proline-rich domain ('PRD', blue), as well as a short, 10-amino acid C terminal tail ('C', white). Variants were generated by deleting the regions flanking the polyQ domain. An additional C-terminal S tag (not shown) was used for immunoblot detection of recombinantly produced mHtt. For recombinant expression of all mHtt variants, a 51-mer polyQ tract was used. (**B**) Kinetics of formation of SDS-insoluble, heat-stable aggregates for the mHtt-Ex1 variants as measured by the filter trap assay. Aggregation of purified recombinant mHtt-Ex1 variants from (**A**) was initiated by cleavage of a solubilizing N-terminal GST tag by TEV protease. All aggregation reactions were performed at a concentration of 3 µM. Data is representative of at least three independent experiments. (**C**) Rate of accumulation of amyloid aggregates for the mHtt variants measured by the ThioflavinT fluorescence assay. All variants were aggregated at a concentration of 3 µM. (**D**) Normalized curves of ThioflavinT amyloid aggregation kinetics from (**C**) used to compare kinetic rates. Variants without the N17 region (ΔN, ΔNΔP) form amyloids much more slowly than variants with the N17 region (Ex1, ΔP). (**E**) Fluorescence images of each of the mHtt mutants transfected into the ST14a striatal neuron-derived cell line. mHtt variants were constructed similar as in (**A**) with a 51-mer polyQ length. Instead of a N-terminal GST and C-terminal S-tag, constructs had only a C-terminal GFP tag. Images were taken 48 hr post transfection. Scale bar is 20 µm. (**F**) Percentage of transfected cells containing aggregates for each of the Htt mutants as shown in (**E**). Similar to as seen in vitro deletion of the N17 region leads to overall less aggregation while deletion of the PRD leads to overall more aggregation. Data are mean ± SEM of three independent experiments counting at least 150 cells. *p <0.05, **p<0.005, ***p<0.001. (**G**) Summary model for how the N17 and PRD regions contribute to mHtt aggregation propensity.

The following figure supplement is available for figure 1:

**Figure supplement 1.** Additional data and modeling of aggregation kinetics.

derived ST14a cells (*Cattaneo and Conti, 1998*). Formation of GFP-inclusions provided a read-out for the aggregation propensity of the mHtt variants in vivo. As observed in vitro, deleting N17 reduced the formation of visible inclusions in vivo, while deleting PRD enhanced the formation of aggregates (*Figure 1E–F*). Notably, few aggregates were visible in the ΔNΔP expressing cells, despite rapid formation of insoluble aggregates in vitro. Given the slow kinetics of amyloid aggregation by ΔNΔP in vitro, it is possible that in the absence of the N17 and PRD flanking regions, the polyQ tract does not efficiently generate amyloidogenic fibrils but instead forms non-amyloidogenic aggregates that are less stable in vivo (*Crick et al., 2013*). We conclude that N17 and PRD have opposing effects of on amyloid formation and aggregation in vitro and in vivo (*Figure 1G*) and further suggest that the cellular environment destabilizes the non-amyloid aggregates generated by the polyQ tract in ΔNΔP.

## N17 and PRD control the morphology of mHtt amyloid fibrils

Next, we used cryo-electron microscopy (cryo-EM) to gain a structural understanding of how N17 and PRD impact the formation of mHtt amyloid fibrils. mHtt-Ex1 fibrils have a characteristic architecture, in which frayed fibril ends branch out from a bundled central core (*Figure 2A*, *Figure 2—figure supplement 1A*) (*Bugg et al., 2012*; *Darrow et al., 2015*; *Shahmoradian et al., 2013*). For the ΔN mHtt variant, we observed dramatically fewer fibrils, consistent with its lower amyloid aggregation propensity (*Figure 1*). In addition, the fibrils formed by ΔN had a strikingly distinct morphology, which lacked the bundled architecture of Ex1 fibrils and were much thinner and straighter (*Figure 2—figure supplement 2*). Allowing ΔN aggregation to reach saturation by prolonged incubation increased the number of fibrils but did not change their thin morphology (*Figure 1C*, *Figure 2—figure supplement 1B*). Thus, the thin fibril structure of ΔN aggregates is intrinsic to the mutation. In contrast, ΔP formed many large, densely packed aggregates with individual fibrils arranged in parallel bundles (*Figure 2A*, *Figure 2—figure supplement 2*), consistent with its increased aggregation propensity. As observed for kinetic measurements, the morphology of ΔNΔP aggregates combined properties from both the ΔN and ΔP fibrils. Similar to ΔN fibrils, the ΔNΔP fibrils were shorter, thinner and lacked the frayed fibril ends observed for Ex1 (*Figure 2—figure supplement 2*); similar to ΔP fibrils, ΔNΔP aggregates consisted of more densely packed fibrils (*Figure 2A*). Quantification of at least 10 individual micrographs for each fibril variant supported these observations, indicating that ΔN fibrils were only several nanometers in width, whereas Ex1 and ΔP fibrils were on average almost a micron wide and over a micron long (*Figure 2—figure supplement 2*). We conclude that N17 and PRD have independent and dramatic effects on the amyloid formation propensity of the polyQ tract and its fibrillar structure. N17 promotes amyloid formation but also enhances interfibrillar contacts, thereby driving the bundling of individual fibrils observed for Ex1 and ΔP (*Figure 2B*). In contrast, PRD appears to destabilize lateral contacts among fibrils and thus prevents their dense packing (*Figure 2B*). The combined effects of N17 and PRD in Ex1 lead to the characteristic mHtt structure consisting of fibril bundles with frayed ends (*Figure 2C*). When N17 is absent, PRD leads to thin sparse fibrils observed for ΔN; while when PRD is absent, N17 leads to thick, bundled aggregates as observed for ΔP (*Figure 2C*).

## The PRD domain controls mechanical stability of mHtt amyloid aggregates

Our model predicts that PRD destabilizes the interfibrillar contacts within an aggregate. This effect may be of importance for HD in light of the prion hypothesis that postulates that intercellular transmission of aggregate 'seeds' can nucleate aggregation in naïve cells (*Pearce et al., 2015*). For prions, distinct fibrillar amyloid stability is a hallmark of different prion strains (*Tanaka et al., 2006*) as their reduced stability (or increased 'frangibility') is directly linked to their propensity for intercellular transmission (*Cushman et al., 2010*). We thus tested if the polyQ flanking regions change the mechanical stability of the mHtt amyloid aggregates. Amyloid aggregates of all mHtt variants were gently isolated by centrifugation and subjected to different sonication conditions to test their stability to mechanical disruption (*Figure 3A*, *Figure 3—figure supplement 1A*). The size and morphology of the resulting species were measured by Dynamic Light Scattering (DLS) and cryoEM, respectively (*Figure 3B–C*, *Figure 3—figure supplement 1B*).

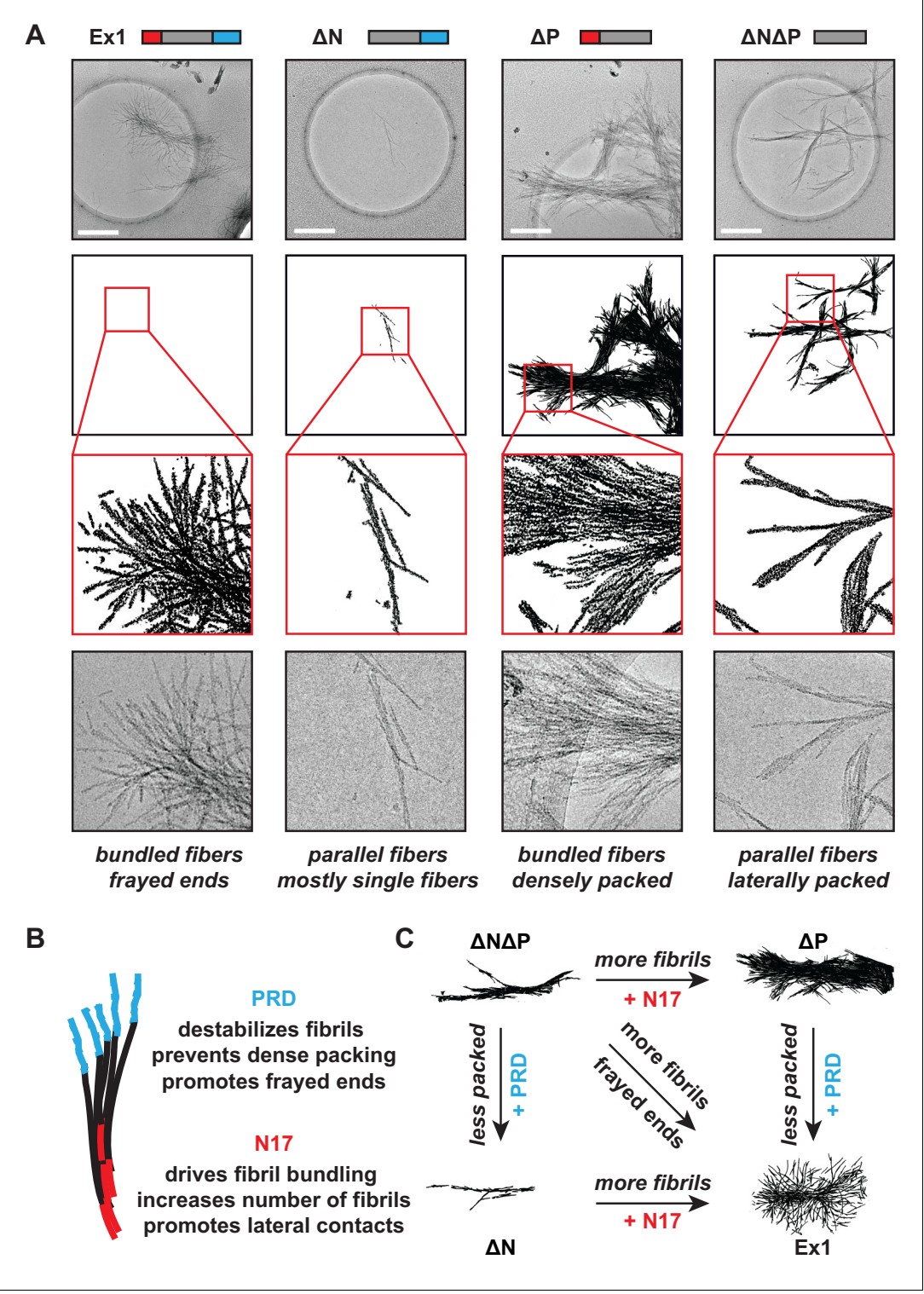

**Figure 2.** Morphology of mHtt aggregation highlight how N17 and PRD contribute to mHtt aggregation. (**A**) Morphology of fiber aggregates from each of the mHtt mutants by cryo-EM. mHtt fibers were imaged 24 hr post initiation of aggregation. Scale bar is 500 nm. Top row is the cryo-EM micrographs. The second and third rows are traced annotations of the EM micrographs and zoom-ins of individual areas, respectively. Bottom row is the original cryo-EM micrograph of the zoomed-in area. (**B**) Summary of the N17 and PRD contributions to aggregation morphology and propensity. (**C**) Summary model for how the N17 and PRD regions direct

*Figure 2 continued on next page*

*Figure 2 continued*

morphology of mHtt aggregates. Morphology differences between ΔNΔP and Ex1 showcase how the flanking regions impact mHtt aggregation in a combinatorial manner.

The following figure supplements are available for figure 2:

**Figure supplement 1.** Additional cryo-EM images of mHtt variant aggregate morphology.

**Figure supplement 2.** Quantification of mHtt variant aggregate morphology.

DLS analyses confirmed that PRD significantly reduced the mechanical stability of mHtt aggregates. mHtt fibrils formed by variants containing a PRD, namely Ex1 and ΔN, were readily fragmented into small, relatively homogeneous seeds of 30–50 nm in size (*Figure 3B–C*). In contrast,

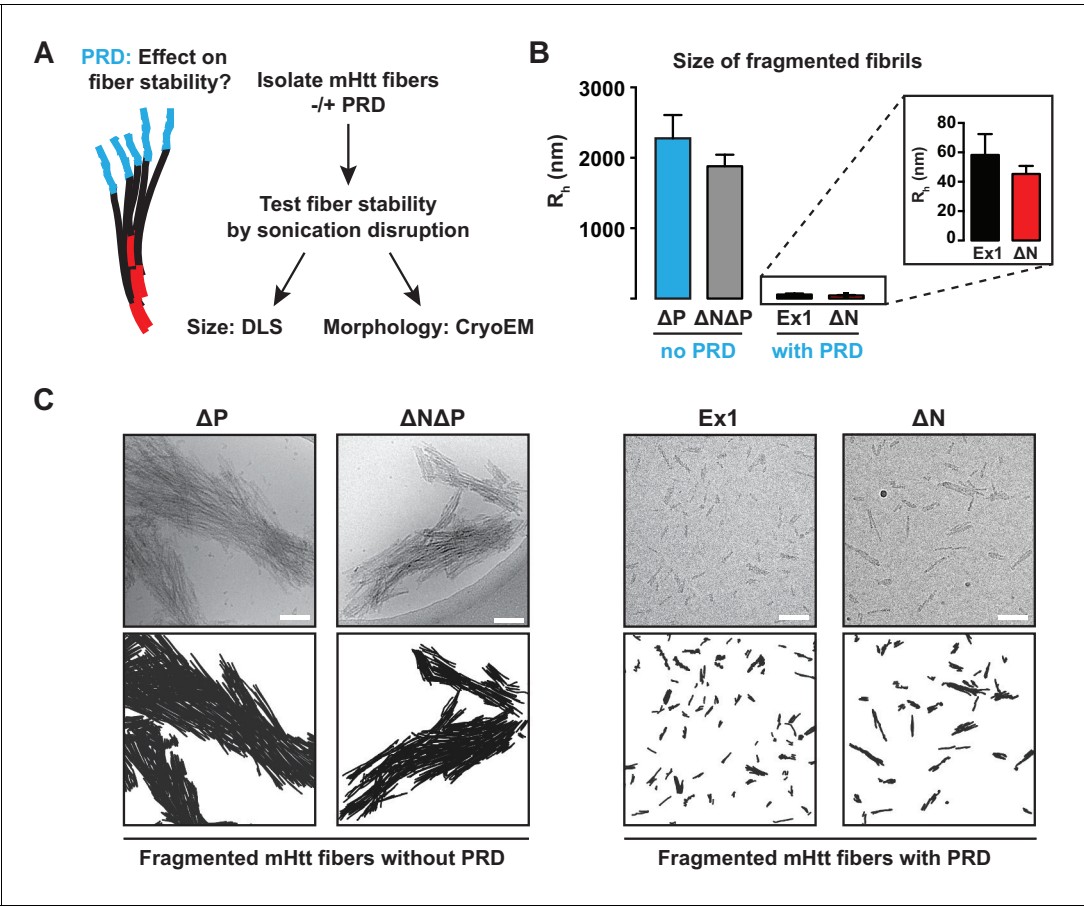

**Figure 3.** The PRD region mechanically destabilizes mHtt aggregates. (A) mHtt fibers were isolated by centrifugation and sonicated to test their mechanical stability. Sonicated species were analyzed by cryo-EM imaging and Dynamic Light Scattering (DLS). (B) Size comparisons between sonicated mHtt aggregates lacking PRD (ΔNΔP, ΔP) and mHtt variants with the PRD (Ex1, ΔN) as measured by DLS. Sonication of mHtt variants containing the PRD results in much smaller fiber fragments than those lacking PRD. Data are mean ± SEM. (C) Cryo-EM images of sonicated fibers for mHtt variants lacking PRD (ΔNΔP, ΔP) or mHtt variants with the PRD (Ex1, ΔN). Scale bar is 100 nm.

The following figure supplements are available for figure 3:

**Figure supplement 1.** DLS data of sonicated mHtt variants at harsher conditions.

**Figure supplement 2.** Comparison of Ex1 and ΔP susceptibility to urea and formic acid denaturation.

aggregates formed by mHtt variants lacking a PRD, namely ∆P and ∆N∆P, resisted disruption and remained in large and highly bundled fibrillar structures ranging from 0.3–2 µm even at the harshest sonication conditions (*Figure 3B–C*, *Figure 3—figure supplement 1*). Cryo-EM imaging confirmed and extended these observations (*Figure 3C*). The sonicated seeds of Ex1 and ∆N were small structures only one or two fibrils in width, while the sonication products of ∆P and ∆N∆P were large, thick, and densely-packed bundles of fibers. Of note, the presence or absence of N17 had no significant impact on the size of the fragmented species, indicating that while N17 contributes to the formation of these fibrils, it does not contribute to their mechanical stability. We conclude that the PRD mechanically destabilizes mHtt aggregates by disfavoring inter-fibrillar contacts (*Figure 2A*), thus providing a molecular basis for reduced aggregation propensity of mHtt variants with a PRD (*Figure 1*).

We next examined whether the PRD domain also contributed to the chemical stability of mHtt fibrils to protein denaturing agents, such as urea or formic acid. The isolated amyloid aggregates of Ex1 and ∆P were initially treated with 8M urea, which was unable to disaggregate either fibril variant (*Figure 3—figure supplement 2A–B*). We then subjected urea-treated fibrils with increasing concentrations of the much harsher denaturant formic acid (*Figure 3—figure supplement 2A–B*). Both Ex1 and ∆P fibrils were similarly resistant to intermediate concentrations of formic acid and only following incubation with 100% formic acid, which can dissolve mHtt aggregates formed in vivo (*Hazeki et al., 2000*), did we observe significant denaturation of the mHtt fibrils of both Ex1 and ∆P (*Figure 3—figure supplement 2B–C*). These experiments suggest that the mechanical stability differences for fibrils with or without the PRD derives from their structural architecture rather than intrinsic differences in their chemical stability.

## N17 and PRD regions shape the conformational landscape of oligomeric mHtt species

We next examined how the polyQ flanking regions impact the conformational ensemble of soluble, oligomeric mHtt species by monitoring the kinetics of formation and stability of soluble, oligomeric species by mHtt flanking region variants (*Haass and Selkoe, 2007*; *Nucifora et al., 2012*; *Sontag et al., 2012*) (*Figure 4A*). At different times following TEV cleavage of the GST moiety, agarose gels (AGEs) were used to examine the formation of oligomeric species larger than 400 kDa (*Figure 4B–C*) (*Sontag et al., 2012*; *Weiss et al., 2008*) while Blue Native acrylamide gels (Blue-Native PAGE) were used to monitor species smaller than 400 kDa (*Figure 4D*). Oligomeric species larger than 400 kDa were resolved under native (*Figure 4B*, *Figure 4—figure supplement 1A*) or mildly denaturing (0.1% SDS) conditions (*Figure 4C*, *Figure 4—figure supplement 1B*). At each time-point, the filter trap assay was used to assess formation of large SDS-insoluble, amyloid aggregates (*Figure 4C*, filter trap).

Native-AGE gel analyses revealed that native oligomers of Ex1 appeared early during aggregation kinetics and gradually decreased as aggregation proceeded to form the aggregates retained in the filter trap (*Figure 4B–C*) (*Sontag et al., 2012*). In contrast, large SDS-soluble oligomers appeared to increase over time in the SDS-AGE gel (*Figure 4C*). By contrast, ∆P had much fewer oligomers in both native- and SDS-AGE gels. Furthermore, the native oligomers disappeared much more rapidly during the time-course of aggregation that those observed for Ex1, consistent with the overall faster aggregation of ∆P. Thus, the absence of PRD alters the balance between oligomer and fibril formation, perhaps by allowing faster progression of these oligomeric species to aggregates (*Figure 1B-C*, *4D*).

Surprisingly, very large ∆N oligomers persisted through the aggregation reaction in both the native-AGE and SDS-AGE gels (*Figure 4B–C*). Even after 60 hr, when ∆N amyloid aggregation was saturated, the population of ∆N oligomers remained under both native and mildly denaturing conditions (*Figure 4—figure supplement 1B*). Consistent with our previous data, the increased presence of ∆N oligomers resulted in few filter-trap retained aggregates (*Figure 1B–C*, *4C*). We wondered if these stable, large ∆N oligomers corresponded to the small, thin ∆N amyloid fibrils observed by cryo-EM (*Figure 2A*). However, while large ∆N oligomers were already abundant as early as 3 hr post-aggregation (*Figure 4B–C*), cryo-EM imaging of ∆N at 3 hr of aggregation showed very few fibrils and no additional identifiable densities or structures (not shown). Furthermore the amyloidogenic, ThioflavinT-reactive species formed by ∆N only accumulated significantly after 30 hr of aggregation (*Figure 1C*). Together, these data indicate that ∆N becomes trapped forming stable soluble

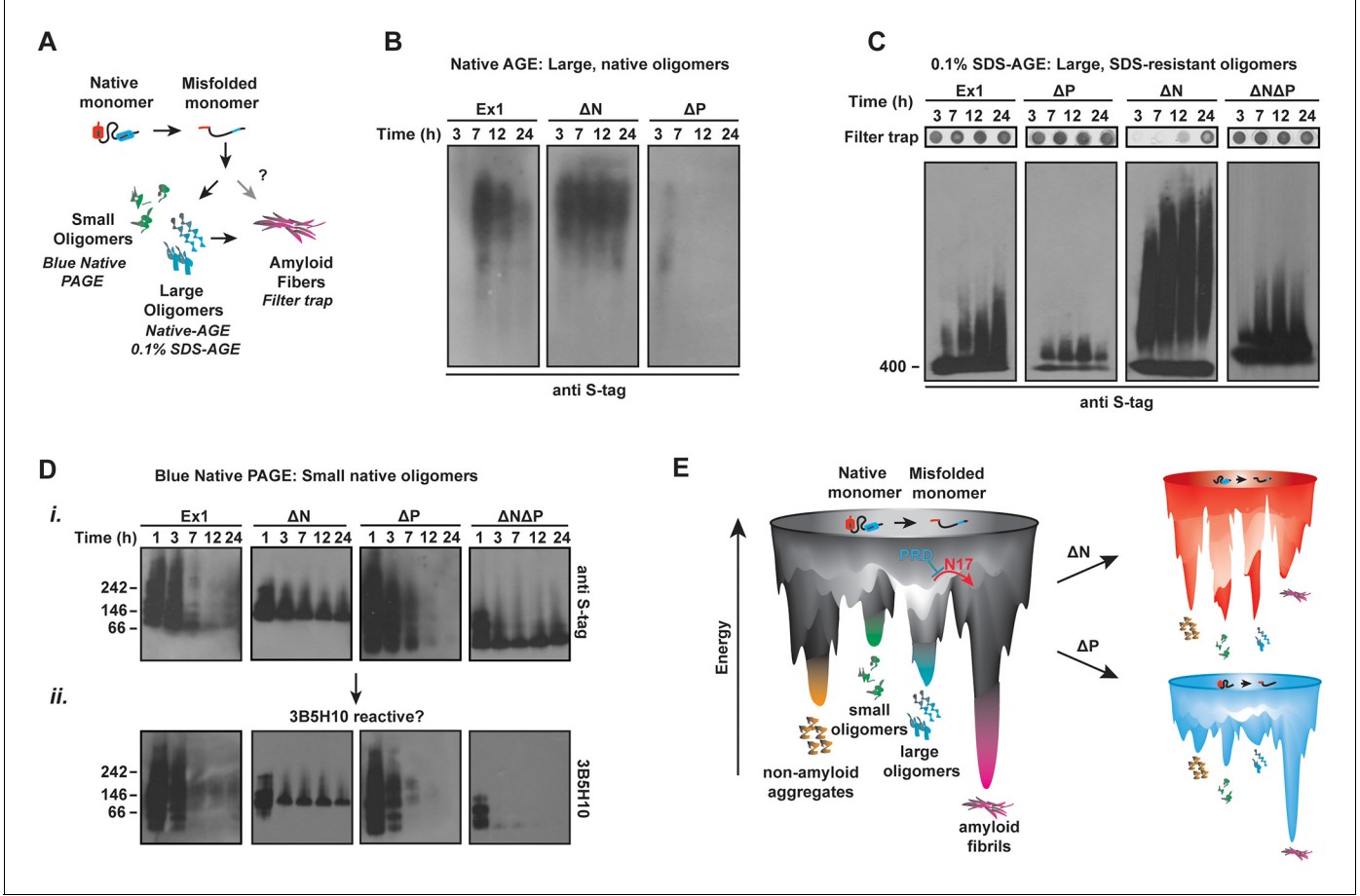

**Figure 4.** N17 and PRD direct the formation and stabilization of oligomer populations. (**A**) Schematic of oligomer and aggregate populations generated during the mHtt aggregation pathway. We characterized the oligomer populations using Agarose Gel Electrophoresis (AGE) under either native or mildly denaturing, 0.1% SDS conditions or Blue-Native PAGE under native conditions. Oligomers were isolated by taking time-points of the in vitro mHtt aggregation reaction, conducted at the same conditions as in *Figure 1*. (**B**) Native-AGE showing native, large oligomers for the mHtt variants. ΔN generated a large population of mHtt oligomeric species that persisted through the aggregation reaction, whereas ΔP generated very few oligomeric species that disappeared quickly from the gel. Blot was immunoprobed for the C-terminal S-tag on the mHtt variants. (**C**) 0.1% SDS-AGE gel showing partially SDS-soluble, oligomers over 400 kDa. Blot was immunoprobed for the C-terminal S-tag on the mHtt variants. Large, SDS-insoluble mHtt aggregates do not enter the SDS-AGE gel, as shown by the reference filter trap (top row, filter trap). ΔN generates few aggregates because the protein remains trapped in the oligomeric range. (**D**) Blue-Native PAGE gel showing native, small oligomers below 400 kDa. Aggregation time-points were run in a Blue-Native PAGE gel and immunoprobed for the S-tag (*i*) and the 3B5H10 conformational antibody (*ii*). 3B5H10 only recognizes the smaller oligomers of the mHtt variants Ex1, ΔN, and ΔP. (**E**) Energy landscape model of the different conformational species in the mHtt-Ex1 aggregation pathway (left). Aggregation through the ΔN or ΔP pathways changes the energetic barriers between the oligomeric and fibrillar states of mHtt (right).

The following figure supplement is available for figure 4:

**Figure supplement 1.** Additional 0.1% SDS-AGE and Native-AGE assays of mHtt-Ex1 variants.

oligomers that may be mildly SDS-resistant but are neither fibrillar nor amyloid. These ΔN oligomers, despite being very large in size, appear to be too structurally heterogeneous to yield identifiable diffraction signatures by cryoEM.

We next examined the impact of the polyQ flanking regions on the formation of soluble oligomer populations smaller than 400 kDa (*Nucifora et al., 2012*; *Schagger and von Jagow, 1991*) (*Figure 4D*). The aggregation reaction was initiated by TEV cleavage of the GST-moieties and time-points analyzed by Blue-Native PAGE (*Figure 4Di*). Interestingly, all mHtt variants formed several small oligomer species very early in the aggregation reaction. For Ex1 and ΔP, these small oligomers

disappeared as aggregation progressed, while the species of ΔN and ΔNΔP were very stable, similar to that observed for large oligomers in SDS-AGE gels (*Figure 4C*). This analysis supports the view that N17 promotes the formation of fibrils (*Figure 2*) at the expense of oligomer formation, whereas the PRD contributes to the accumulation of oligomers.

Interestingly, N17 and PRD seem to act independently in oligomer formation since the oligomers formed by ΔNΔP exhibited a behavior intermediate between the ΔN and ΔP phenotypes in the AGE oligomer assays. While a significant amount of ΔNΔP species progressed to the insoluble aggregates (*Figure 4*, filter trap), a population of trapped oligomers persisted in the SDS-AGE gel and in the Blue-Native PAGE gel. It thus appears that N17 and PRD have independent and antagonistic effects on the conformational landscape of the polyQ (*Figure 4E*). N17 promotes amyloid formation and fibril bundling and prevents the accumulation of non-amyloidogenic oligomeric species. On the other hand, PRD disfavors aggregation by enhancing the levels of soluble oligomeric species while structurally destabilizing the amyloid fibril association.

We next considered whether the flanking regions also influence the structural properties of the oligomers, as observed for the fibrils. Indeed, their distinct migration patterns in Native-AGE and Blue-Native PAGE, which separate oligomers based on a number of shape, size and exposed charge characteristics, seems to support this hypothesis. Unfortunately, characterizing these oligomeric species by conventional structural approaches such as electron microscopy, crystallography or NMR is technically challenging given that these oligomeric species are structurally heterogeneous and chemically unstable. Conformational antibodies have provided a useful approach to distinguish specific conformations of amyloidogenic proteins, albeit the characteristics of these reactive species are not often defined (*Brooks et al., 2004*; *Kayed and Glabe, 2006*; *Kayed et al., 2003*). These conformational antibodies have also been proposed to recognize more toxic conformations of amyloidogenic species (*Kayed et al., 2003*; *Miller et al., 2011*). We tried a panel of conformationally-sensitive antibodies to identify conformational subsets in the oligomer populations separated in our AGE and PAGE analyses. We failed to observe significant reactivity of the oligomers of any flanking domain variant with A11 or OC conformationally-sensitive antibodies (not shown). Interestingly, we did observe reactivity with the 3B5H10 antibody, previously proposed to recognize a toxic polyQ conformation (*Miller et al., 2011*). The 3B5H10 reactive species were small oligomers resolved by the Blue-Native PAGE gels, while none of the larger oligomers in the SDS-AGE gel were 3B5H10 reactive (*Figure 4—figure supplement 1C*) (*Miller et al., 2011*).

Interestingly, 3B5H10 reactivity did support the hypothesis that the flanking regions influence the structural conformation of the oligomers. We observed that Ex1, ΔN and ΔP all formed small 3B5H10-reactive oligomers, albeit of different sizes and mobilities. In contrast, the small oligomers formed by ΔNΔP, which had similar mobility to those formed by ΔN, were not 3B5H10 reactive (*Figure 4Dii*). This suggests that the conformation adopted by the polyQ tract within oligomers is influenced by the flanking regions. Thus, the flanking regions influence structural differences in not only the fibrillar aggregate species but also in soluble species affecting both the stabilities and conformations of the heterogeneous species formed along the mHtt aggregation pathway.

## N17 triggers a molecular switch from oligomers to fibrils

In considering the relationship between the polyQ flanking regions and oligomer populations, the above results suggest that N17 plays a role in promoting the oligomer to fibril transition (*Figure 5A*). Indeed, previous studies showed that *trans* addition of an N17 peptide to ΔN or even to Ex1 enhanced formation of filter-trappable aggregates (*Tam et al., 2009*). We thus tested if N17 acts by converting 'kinetically trapped' ΔN oligomers to an amyloid-competent conformation that proceeds to fibrillar aggregates. To this end, we added the N17 peptide *in trans* to a ΔN aggregation reaction and examined the aggregates by cryoEM (*Figure 5A–B*). The ΔN fibers formed without N17 peptide were sparse, thin, and short, as expected (*Figure 5B*) and no aggregates were observed for N17 alone (not shown). Strikingly, we observed that *trans* addition of the N17 peptide led to a remarkable increase in the amount of ΔN fibrils (*Figure 5B*). Furthermore, the N17-treated ΔN fibrils exhibited the distinctive bundled morphology observed for Ex1 (*Figure 5B*, *Figure 5—figure supplement 1A*, *Figure 2A*).

The effect of *trans* addition of N17 on the trapped ΔN oligomer populations was next examined by the addition of two different N17 peptide concentrations to a ΔN aggregation reaction. Oligomer formation was assessed by SDS-AGE and immunoblotting against ΔN (*Figure 5C*). The addition of

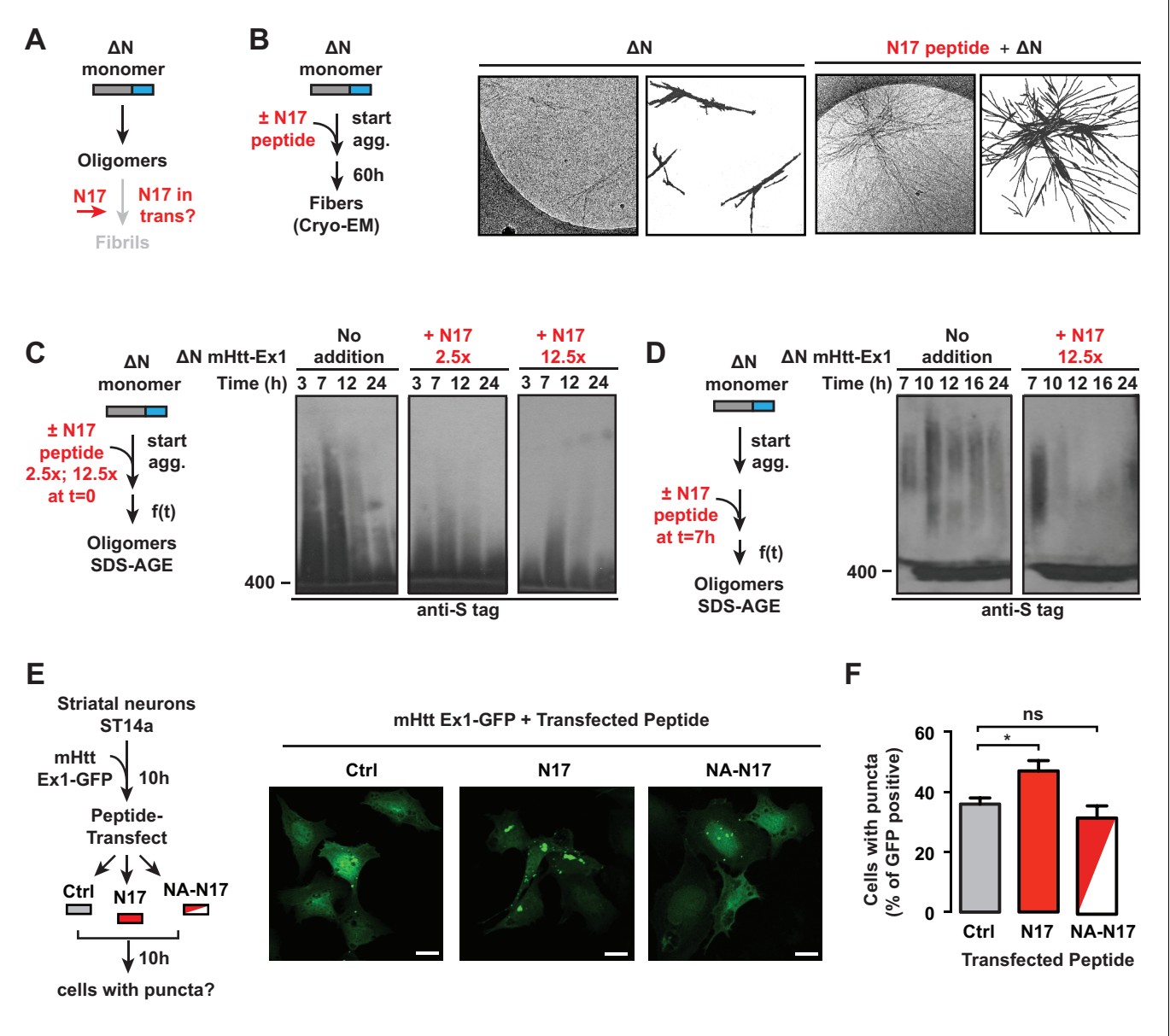

**Figure 5.** N17 is essential to promote transition of oligomers to amyloid aggregates. (**A**) General schematic of *trans* N17 addition experiments to ΔN aggregation reaction to determine how N17 impacts the balance between stable, ΔN oligomers and amyloid fibrillar aggregates. (**B**) 12.5x excess of N17 peptide was added when ΔN aggregation was initiated. Then 60 hr after initiation of aggregation, the fibers were analyzed by cryo-EM. N17-seeded ΔN aggregates become more bundled and more resemble the morphology of Ex1 aggregates (*Figure 2A*). (**C**) 12.5x or 2.5x excess of N17 peptide was added *in trans* when ΔN aggregation was initiated. The impact of N17 peptide on ΔN oligomers was analyzed by 0.1% SDS-AGE gels and immunoprobed for the C-terminal S-tag. *Trans* addition of N17 promotes disappearance of these stable ΔN oligomers from the AGE gel. (**D**) 12.5x excess of N17 peptide was added *in trans* 7 hr after initiation of ΔN aggregation, allowing for the formation of ΔN oligomers before N17 addition. Impact of N17 peptide on ΔN oligomers was analyzed by 0.1% SDS-AGE gels and immunoprobed for the C-terminal S-tag. (**E**) Schematic of in vivo N17 addition experiment: ST14a striatal-derived neurons were transfected with the C-terminally GFP tagged mHtt-Ex1 (mHtt-Ex1-GFP). 10 hr after transfection, when expressed mHtt protein is still completely soluble, cells are protein transfected using the Xfect kit (Clontech) with N17 or a mutant N17 peptide (NA) that inhibits aggregation. After 10 hr, resulting cells containing GFP fluorescent puncta are counted. (**F**) Fluorescence microscopy of ST14a striatal neurons transfected with mHtt-Ex1-GFP and N17 variant peptides (left) and quantification of cells containing puncta (right). Data are mean ± SEM of three independent experiments with at least 200 cells counted in each condition. Scale bar is 20 μm. *p<0.05.

The following figure supplement is available for figure 5:

**Figure supplement 1.** Additional cryo-EM images of ΔN aggregates with N17 peptide added *in trans* and transfection of N17-peptide into neurons expressing Htt-Q25.

N17 at the start of the aggregation reaction led to the disappearance of ΔN oligomers (*Figure 5C*), consistent with the increase in aggregates observed by EM and filter trap (*Figure 5B*) (*Tam et al., 2009*). The rate of oligomer disappearance was enhanced at higher N17 concentrations (*Figure 5C*). Thus, it seems that N17 triggers a conformational switch that disfavors the accumulation of trapped, oligomeric species and allows the formation of amyloid fibrils. Such a role explains the slow ΔN amyloid formation observed in the kinetic and cryo-EM analyses (*Figure 1B–C*, *Figure 2A*). We next asked whether N17 could impact pre-formed, stable ΔN oligomers. We incubated ΔN for 7 hr to generate a large population of oligomers but no fibrils, then added N17 *in trans* (*Figure 5D*). Strikingly, N17 also promoted disappearance of these pre-formed, non-amyloidogenic oligomers (*Figure 5D*), suggesting that N17 can act on the kinetically trapped ΔN oligomers themselves to promote formation of amyloid fibrils (*Figure 5B*).

The ability of the N17 peptide to modulate aggregation in vivo was next examined following mHtt-ex1-GFP transfection in ST14a striatal-like neurons (*Figure 5E*). At 10 hr post-mHtt transfection, when mHtt-Ex1 is still diffusely localized in all cells, we transfected the cells with either the N17 peptide or an N17 peptide mutant that cannot promote aggregation in vitro ('NA-N17') (*Tam et al., 2009*) (*Figure 5E*). We next followed the formation of GFP-positive aggregates in these cells versus a buffer control. While no significant differences were observed between Ctrl-treated and NA-N17 treated cells, we observed a significant increase in the number of cells with mHtt-GFP inclusions in the N17-transfected cells (*Figure 5F*). Of note, N17 did not promote inclusion formation in ST14a cells expressing the non-pathogenic Q-length Htt-Ex1-Q25-YFP (*Figure 5—figure supplement 1B*). These results indicate that, N17 *trans* addition can increase mHtt aggregation in vivo.

## Relevance of fibrillar aggregates versus oligomers in mHtt neuronal toxicity

Since N17 and PRD exert opposing effects on the mHtt conformational landscape, they present an opportunity to test the competing models that neuronal toxicity is caused by the formation of amyloid aggregates or the formation of oligomers. Specifically, ΔN forms many oligomers and few fibrillar aggregates; ΔP forms many fibrillar aggregates and few oligomers; and ΔNΔP, which is essentially a pathogenic length Q-tract, forms both oligomers and fibrils (*Figure 6A*). To evaluate the toxicity of these polyQ expanded mHtt variants we looked at the striatal neurons of rat corticostriatal brain slices, which provide a more disease relevant, 'tissue contextual' model of HD when compared to traditional models of neuronal cell cultures (*Khoshnan et al., 2004*; *Reinhart et al., 2011*). In corticostriatal brain slices, the striatal medium spiny neurons (MSNs), which are preferentially affected in HD, remain within their intact local tissue environment and maintain interactions among multiple brain cell-types. mHtt toxicity was measured by co-transfecting corticostriatal brain slices with a plasmid encoding untagged mHtt-Ex1 variants and another plasmid encoding YFP, which serves as an independent morphological marker for the transfected neurons (*Crittenden et al., 2010*) (*Figure 6B*). MSN viability is measured by examining the cellular and dendritic morphology by YFP expression (*Figure 6C–E*). Importantly, when corticostriatal brain slices are transfected with polyQ-expanded mHtt there is a progressive degeneration of MSNs over 3–5 days (*Reinhart et al., 2011*). No degeneration is observed when YFP is co-transfected with Htt carrying non-pathogenic polyQ repeats (Q8, Q23) or with a control vector (*Reinhart et al., 2011*).

Surprisingly, when the mHtt-Ex1 variants were transfected into brain slices, there were striking discrepancies between their toxicity and their structural properties. Although ΔN and ΔP have dramatically different aggregation and oligomerization propensities in vitro and in vivo, both were as toxic as Ex1 as early as 3 days after transfection (*Figure 6C–E*). Even more surprisingly, ΔNΔP, which generates both aggregates and oligomers and has an expanded polyQ tract of identical length, exhibited only modest toxicity even by day 4 (*Figure 6E*). These results negate simple models of aggregate-only or oligomer-only toxicity and instead call for a more nuanced view of proteotoxicity. Since both ΔN, which forms few amyloid fibrils, and ΔP, which forms few oligomers are both highly toxic, our data cannot be explained by either fibrillar aggregates or, oligomeric species per se as being toxic. Remarkably, since the expanded polyQ tract itself is only mildly toxic, the action of the flanking regions must be key to induce the toxic polyQ conformation(s).

Our unexpected toxicity findings suggest a number of hypotheses. First, it is possible that there is no universal proteotoxic species, and either specific fibrillar or oligomer states mediate toxicity, most likely through different mechanisms given their radically different properties. However, it was

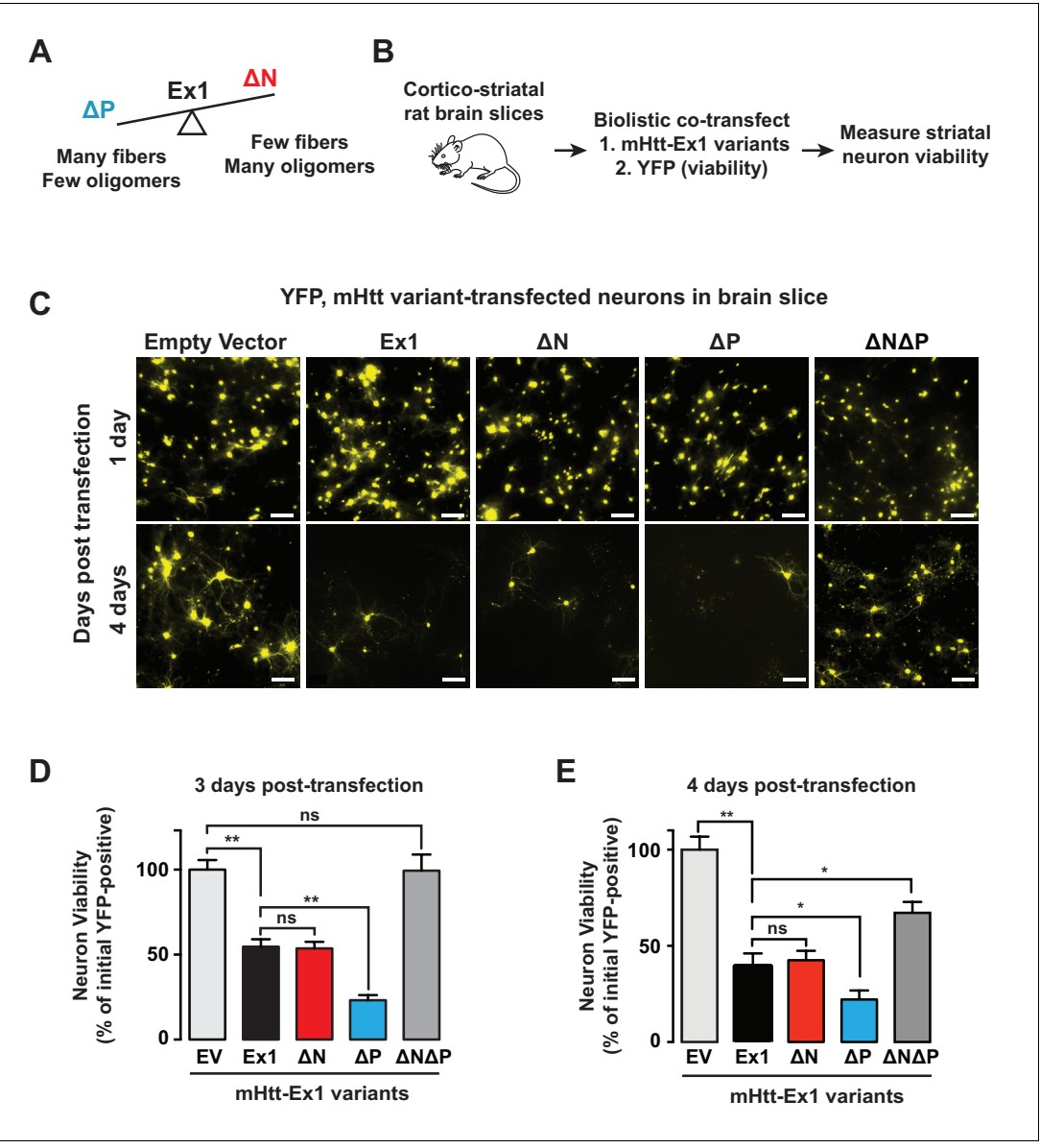

**Figure 6.** Assessing proteotoxicity of mHtt variants in corticostriatal brain slices. (**A**) Compared to Ex1, ΔN forms many more oligomers and much fewer fibrillar aggregates, whereas ΔP forms many more fibrillar aggregates and much fewer oligomers. (**B**) Schematic of toxicity assay: Corticostriatal slices were prepared from rat brains and biolistically co-transfected with mHtt-Ex1 variants or an Empty Vector control (EV) and an independent YFP plasmid as a marker of cell viability. Viability of medium spiny neurons (MSNs) was visually assessed by cell morphology and YFP expression after 3–4 days post-transfection. (**C**) Fluorescence images of MSNs from the brain slices co-transfected with the mHtt variants and YFP. Images were taken 1 day (top row) and 4 days (bottom row) after transfection. Data are representative of at least three independent experiments. Scale bar is 20 μm. (**D**) Relative viability of MSNs transfected with mHtt variants 3 days after transfection. Data are mean ± SEM. Data are representative of at least three independent experiments. **p<0.0001 (**E**) Relative viability of MSNs transfected with mHtt variants 4 days after transfection. Data are mean ± SEM. Data are representative of at least three independent experiments. *p<0.01, **p<0.0001

The following figure supplement is available for figure 6:

**Figure supplement 1.** Transfected Htt-Ex1 is only toxic in transfected MSNs at pathogenic Q-lengths.

intriguing that mHtt toxicity in the brain slices was observed for those variants that gave rise to 3B5H10-reactive small oligomers, namely Ex1, ΔN and ΔP but not ΔNΔP. Thus, an alternative is that toxicity resides in a subclass of structural conformers, such as oligomers linked to 3B5H10 reactivity, rather than a particular size of species. Yet another possibility is that mHtt interconverts between several conformations when expressed in vivo and that toxicity arises when those species cannot be efficiently cleared by the cellular machinery. We thus extended our analysis to the soluble mHtt conformers formed in striatal neurons, as well as their interactions with the proteostasis machinery.

## Influence of N17 and PRD on mHtt conformation in neurons

To understand how the neuronal cellular environment impacts the mHtt conformational ensemble, we tested whether in vivo the 3B5H10-reactive conformation was also specifically observed for the toxic variants Ex1, ΔN and ΔP, but not for the less toxic ΔNΔP (*Figure 7A–B*) (*Brooks et al., 2004*; *Kayed and Glabe, 2006*; *Kayed et al., 2003*). Indeed, immunofluorescence analysis showed that striatal-derived neurons expressing the toxic variants Ex1, ΔN and ΔP contained a diffusely localized 3B5H10-reactive species that was absent in cells expressing the less toxic ΔNΔP (*Figure 7B*). Little 3B5H10 reactivity was observed in neurons expressing Htt-Ex1 construct with a non-pathogenic polyQ length (*Figure 7—figure supplement 1A*), similar to a previous report (*Miller et al., 2011*). Of note, 3B5H10 did not stain large mHtt aggregates or inclusions (*Figure 7C*), suggesting that in vivo the 3B5H10-reactive conformation is excluded from large aggregates.

We further examined the soluble mHtt conformations generated in vivo by subjecting the ST14a neuronal extracts to SDS-AGE and Blue-Native PAGE analyses, followed by GFP and 3B5H10 immunoblotting (*Figure 7D*). GFP immunoblotting of the Blue-Native PAGE revealed variant-specific single bands, suggesting that, unlike what we observed in vitro each variant forms a predominant small oligomeric species in vivo (*Figure 7D*). In contrast, the larger mHtt oligomers observed in vivo displayed a heterogeneous spectrum of sizes as observed with purified mHtt. Of note, the only 3B5H10-reactive species formed in vivo were the small mHtt oligomers detected in the Blue-Native PAGE (*Figure 7D*). None of the larger oligomeric species detected in SDS-AGE reacted with 3B5H10 (*Figure 7—figure supplement 1B*). Notably, as seen in vitro, 3B5H10 only recognized the small oligomers formed by the toxic variants Ex1, ΔN, and ΔP but not the oligomers formed by ΔNΔP (*Figure 7D*).

We conclude that N17 and PRD influence the kinetics and stabilities of oligomers during aggregation and also modulate the toxic conformations adopted by the polyQ stretch in these oligomers. The polyQ tract itself can clearly generate different oligomer populations, but formation of oligomer conformations reactive to 3B5H10 requires either the N17 or PRD domain. The similar 3B5H10 reactivity of Ex1, ΔN, ΔP and ΔNΔP in vitro and in vivo suggests that the conformational differences among these mHtt variants is intrinsic to the structural influence of the flanking regions on the polyQ region.

While it is tempting to speculate that these soluble, 3B5H10-reactive small oligomers are by themselves the toxic Htt species, it is possible that additional toxic species exist but are not recognized by any existing conformational antibody. Nonetheless, these experiments demonstrate that mHtt variants that produce neurotoxic species share specific conformers that are disfavored in the ensembles formed by the less toxic, expanded polyQ tract alone.

## Neuroprotective addition of chaperonin ApiCCT1 reduces 3B5H10 reactive oligomer levels

We analyzed how known protective proteostasis mechanisms impact the presence of small oligomers in vivo. One established mechanism is the chaperonin TRiC/CCT, which suppresses mHtt aggregation and toxicity in many HD models through recognition of N17 by the CCT1 subunit (*Behrends et al., 2006*; *Kitamura et al., 2006*; *Tam et al., 2006*; *Tam et al., 2009*). In neuronal cell culture, overexpression of CCT1 or exogenous addition of the purified substrate-binding domain of CCT1 (ApiCCT1) to the media suffices to suppress mHtt toxicity (*Kitamura et al., 2006*; *Sontag et al., 2013*; *Tam et al., 2009*). We thus examined if ApiCCT1 could also be neuroprotective in the corticostriatal brain slice toxicity model of HD (*Figure 7E*). Indeed, exogenous administration of purified recombinant ApiCCT1 significantly protected MSNs from mHtt neurotoxicity in a concentration-dependent manner (*Figure 7Ei*). Strikingly, Blue-Native PAGE revealed that this exogenous

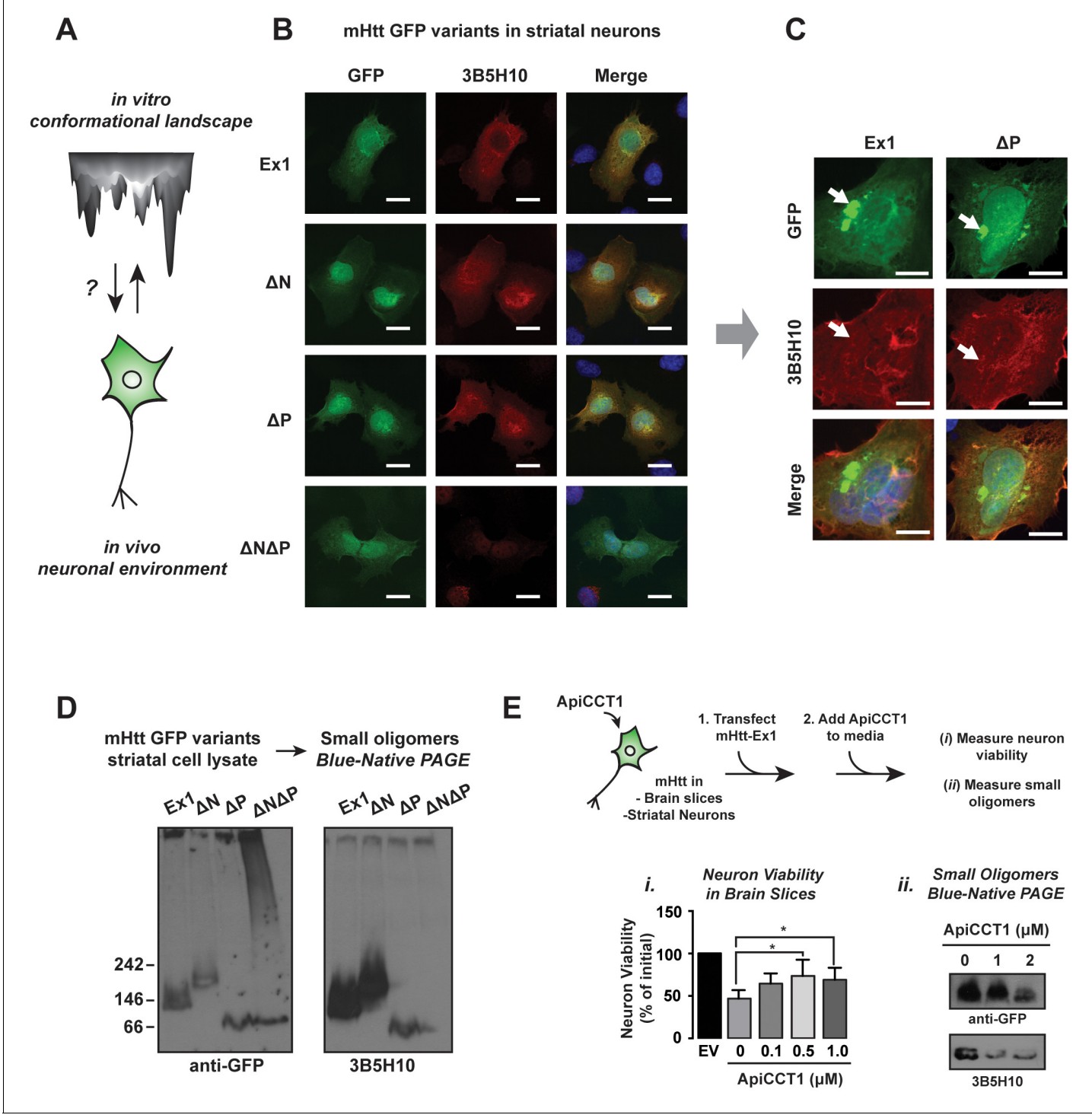

**Figure 7.** Impact of PolyQ flanking domains of the formation of soluble oligomers in vivo. (**A**) We investigate the impact of the neuronal environment on the mHtt conformational landscape. (**B**) Confocal imaging of ST14a striatal-derived neurons transfected with mHtt-Ex1 variants C-terminally tagged with GFP. Cells were imaged in the GFP channel well as immunostained with the 3B5H10 polyQ conformational antibody. Scale bar is 20 μm. (**C**) Confocal images of Ex1 and ΔP-transfected ST14a cells containing puncta. Cells were imaged in the GFP channel as well as immunostained with the 3B5H10 polyQ conformational antibody. The 3B5H10 antibody shows no specific recognition of the GFP puncta. Scale bar is 10 μm. (**D**) Immunoblots of lysates from ST14a striatal neurons transfected with mHtt-Ex1 variants C-terminally tagged with GFP and run in Blue-Native PAGE gels. Blots were probed for GFP (left) and the 3B5H10 conformational antibody (right). (**E**) Schematic for ApiCCT1 addition experiment: Corticostriatal brain slices (*i*) and ST14a striatal-derived neurons (*ii*) were transfected with mHtt-Ex1 simultaneously with exogenous addition of ApiCCT1 to media. Then, viability of

*Figure 7 continued on next page*

*Figure 7 continued*

MSNs in the corticostriatal brain slices was measured (*i*) and 3B5H10-positive small oligomers in the striatal neuron extracts were characterized (*ii*). Data are mean ± SEM. **p<0.01.

The following figure supplement is available for figure 7:

**Figure supplement 1.** Immunofluorescence images and 0.1% SDS-AGE of ST14a cells transfected with Htt-Ex1 variants and immunoprobed with 3B5H10.

ApiCCT1 addition caused a concentration-dependent reduction in the levels of small 3B5H10-reactive oligomers implicated in toxicity (*Figure 7Eii*). These experiments support the notion that these 3B5H10-reactive small oligomers are diagnostic of a conformational pathway linked to toxicity and that chaperonin protection against HD toxicity decreases the levels of these soluble species.

## N17 and PRD act in synergy to control mHtt proteostasis

The toxicity of mHtt populations depends not only on their biophysical properties but also their interactions with cellular proteostasis pathways (*Kim et al., 2016*), which may also depend on the influence of the polyQ flanking domains (*Figure 8A*). We first used corticostriatal brain slices to investigate the impact of the different polyQ flanking domain variants on TRiC-mediated neuroprotection. CCT1 was cotransfected with the toxic mHtt-Ex1 variants into the corticostriatal brain slices and MSN viability assessed as above (*Figure 8B*, *Figure 8—figure supplement 1A*). As expected, overexpression of CCT1 dramatically protected MSNs from mHtt-Ex1 toxicity but did not protect MSNs from ΔN-induced toxicity. These results are consistent with the notion that N17 hosts the CCT1 binding site in mHtt (*Tam et al., 2009*). Surprisingly, despite the presence of N17, CCT1 afforded only a modest degree of protection for ΔP-induced toxicity. The minimal rescue of ΔP by CCT1 may be due to the aggressive neurotoxicity of ΔP (*Figure 6C–E*), which may impair neurons too quickly to be effectively rescued by CCT1. Alternatively, we considered that the absence of the PRD domain may reduce the availability of N17 to engage CCT1.

To further probe the interplay between N17 and PRD in vivo, we examined two additional cellular processes regulated by N17. First, N17 contains a nuclear export sequence (*Gu et al., 2015*; *Maiuri et al., 2013*; *Zheng et al., 2013*), leading ΔN variants to accumulate in the nucleus. In striatal-derived neurons, we confirmed that mHtt-Ex1 was both cytoplasmic and nuclear while ΔN was enriched in the nucleus (*Figure 8C*). Surprisingly, ΔP was also enriched in the nucleus, even though it contains the N17 domain (*Figure 8C*), further suggesting that the PRD region may regulate the exposure of N17.

Second, N17 is also proposed to promote mHtt degradation (*Thompson et al., 2009*), which occurs via the ubiquitin-proteasome and autophagy pathways (*Thompson et al., 2009*; *Tsvetkov et al., 2013*). A pulse of $^{35}$S–methionine labeling followed by a non-radioactive chase measured the impact of N17 and the PRD on the clearance of soluble mHtt variants, which were detected by immunoprecipitation and autoradiography (*Figure 8D*, *Figure 8—figure supplement 1B–E*). The pulse began 14 hr post-transfection when cellular mHtt was still soluble (not shown) to ensure labeling of only soluble mHtt populations. Under these conditions, the disappearance of mHtt-Ex1 is abrogated by inhibition of the proteasome and lysosomal pathways, as expected (not shown). Notably, both ΔN and ΔP had significantly slower protein clearance rates compared to Ex1 (*Figure 8—figure supplement 1A–B*). This suggests that efficient mHtt recognition by the cellular degradation pathways requires the presence of both polyQ-flanking domains.

To clarify the synergy between the PRD and N17, we considered the structural and energetic contributions of the two domains to mHtt conformation. As N17 is critical to drive the initial stages of amyloidogenesis and aggregation and the absence of the PRD promotes rapid aggregation, N17 may become more buried in the faster-forming ΔP oligomer species. We thus assessed the relative exposure of N17 in purified mHtt-Ex1 variants with and without PRD by exploiting the enhanced protease susceptibility of exposed unstructured protein regions. Oligomers formed by Ex1 and ΔP were generated in vitro and the degree of exposure of N17 was tested by a short incubation with low concentrations of Proteinase K, a broad specificity protease (*Figure 8Ei*). N17 was then detected by SDS-PAGE immunoblotting with an anti-N17 antibody (*Figure 8Ei*). We found that N17 was

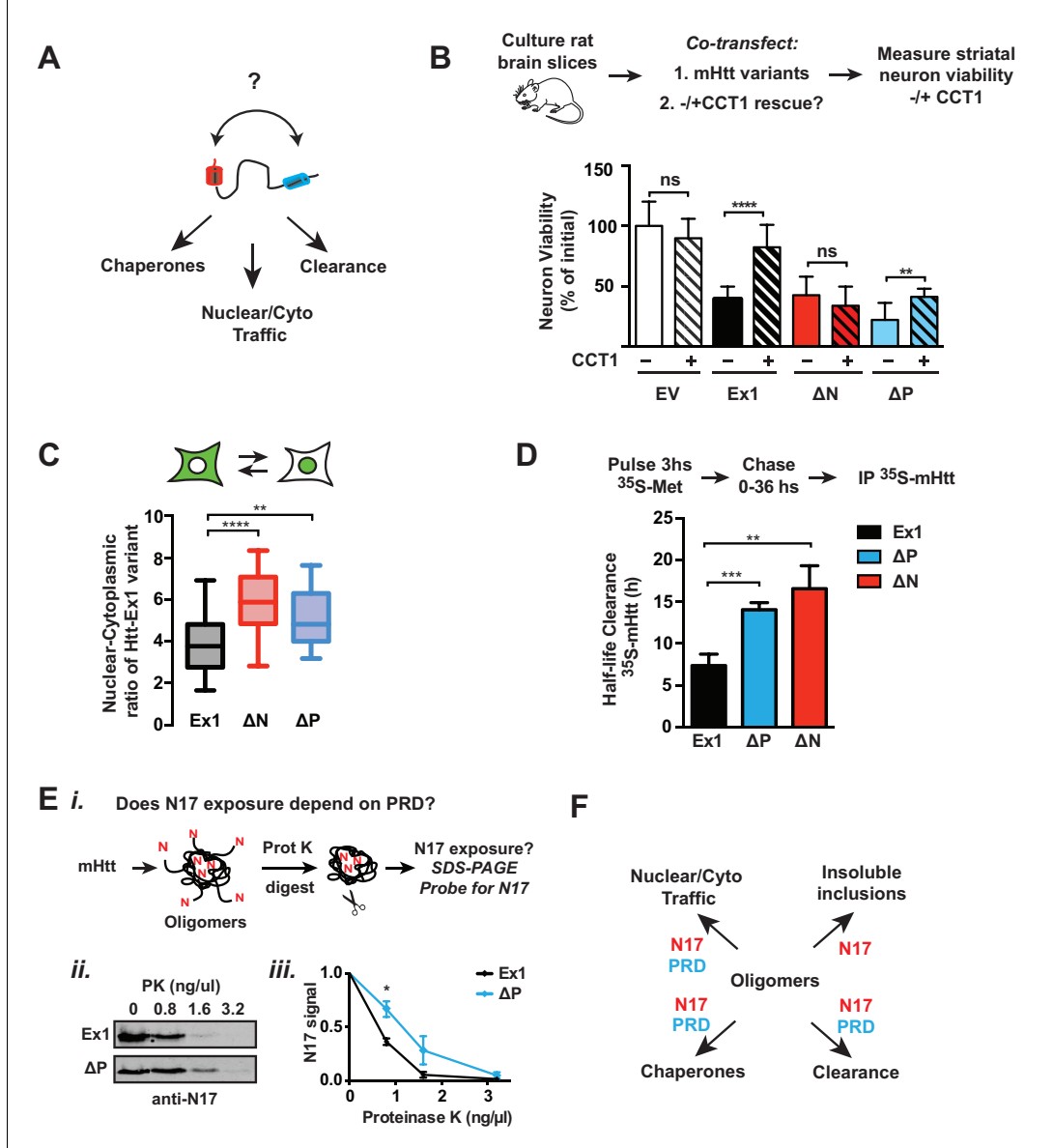

**Figure 8.** Flanking domains act synergistically in vivo to determine mHtt fate. (A) Htt flanking domains engage various cellular proteostasis pathways (B) Relative viability of MSNs from rat brain slices biolistically co-transfected with the toxic mHtt variants, YFP, and the CCT1 subunit of the TRiC chaperonin. Viability was quantified 4 days after transfection. Data are mean ± SEM. Data are representative of at least three independent experiments. ****p<0.0001 (C) Nuclear and cytosolic distribution of GFP-tagged mHtt variants transfected into ST14a neurons. Distribution was quantified by measuring the ratio of mean intensity between nuclear and cytosolic fluorescent signals of individual cells. Data are mean ± SEM with at least 20 cells counted per condition. ****p<0.0001, **p<0.01 (D) $^{35}$S pulse-chase measuring soluble protein degradation of mHtt-Ex1 variants in transfected ST14a cells. After transfection, cells were pulsed with $^{35}$S methionine for 3 hr and chased with complete media. Respective time points were lysed and subject to a clearing spin to only characterize soluble mHtt. Soluble mHtt was immunoprecipitated using the GFP tag. (E) (*i*) Schematic of ProteinaseK (PK) digestion experiment: Ex1 and ΔP oligomers were generated in vitro, then digested with increasing concentrations of ProteinaseK. Protease-digested reactions were run in a SDS PAGE gel and probled for N17. (*ii*) SDS-PAGE gel of ProteinaseK-digested Ex1 and ΔP oligomers immunoprobled for N17. (*iii*) Quantification of the N17 signal from the SDS-PAGE gel in (*ii*). Data are mean ± SEM. *p<0.05. (F) Model for how the polyQ flanking regions contribute to mHtt proteostasis and cellular mechanisms for clearance of toxic species.

The following figure supplement is available for figure 8:

**Figure supplement 1.** Trypsin sensitivity of Ex1 and ΔP oligomers and $^{35}$S pulse-chase measuring protein degradation for mHtt-Ex1 variants.

significantly more protease-sensitive in Ex1 oligomers containing the PRD than in ΔP oligomers (*Figure 8Eii–iii*). As an alternative assay, we exploited the fact that N17 is the only region in mHtt-Ex1 that contains lysine residues, which are specifically cleaved by trypsin. Since Ex1 and ΔP contain the same N17 sequence, any differences in trypsin digestion will arise from the differential exposure of N17. Indeed, N17 in Ex1 oligomers was much more sensitive to trypsin protease cleavage than in ΔP oligomers (*Figure 8—figure supplement 1F*). Thus, these experiments show that the PRD increases exposure of the N17 domain, which explains the similar responses of ΔN and ΔP to CCT1 protection, their enhanced nuclear localization and their slower protein degradation kinetics, all of which were previously shown to depend on N17 (*Figure 8B–D*). Taken together, these analyses indicate that, despite their opposing effects on the energetics of aggregation or oligomer formation, the two polyQ flanking regions also act synergistically in vivo to determine the cellular fate of mHtt (*Figure 8E*).

## Discussion

Despite the link between polyQ-tract length, aggregation propensity and disease severity, the forces shaping the ensemble of mHtt conformational species and their relation to neuronal toxicity remain elusive. Here we define how the conformational landscape of mHtt is controlled by the polyQ-tract flanking domains N17 and PRD (*Figure 4E*). Importantly, the structural influence of these regions and their ability to interact in vivo with proteostasis factors determine the cellular fate of mHtt and the generation of neurotoxic mHtt conformations (*Figure 9*).

### N17 and PRD impact energetics of mHtt conformational landscape

While an expanded polyQ tract is intrinsically aggregation-prone, we find that by itself it inefficiently generates toxic conformations when expressed in MSNs of corticostriatal brain circuits. In vitro the ΔNΔP variant, encompassing essentially only the polyQ tract, accumulates filter-trappable SDS-insoluble species at similar rates as mHtt-Ex1 and also forms large and small oligomers. However, the conformations populated by ΔNΔP are clearly different from those of Ex1: its rate of amyloid formation measured by ThioflavinT is much slower, similar to the toxic ΔN variant and its fibrils are more tightly bundled, similar to the toxic ΔP (*Figure 1B–C*, *Figure 2A*). Supporting the role of flanking domains in generating specific polyQ conformers, ΔNΔP does not give rise to 3B5H10 reactive species, unlike the toxic Ex1 variants, even though this antibody is known to recognize a polyQ-containing β-hairpin conformation (*Brooks et al., 2004*; *Peters-Libeu et al., 2012*). These data suggest that in the absence of flanking regions, the polyQ tract quickly forms large, non-amyloidogenic oligomers and SDS-insoluble aggregates but takes longer to develop an amyloid conformation (*Binette et al., 2016*; *Crick et al., 2013*) and is less likely to populate toxic conformation(s).

N17 and PRD affect the energetic barriers leading to formation of diverse polyQ oligomers and aggregates, as well as their structural characteristics. We propose that N17-N17 and N17-polyQ interactions lower the kinetic barrier between non-amyloidogenic oligomeric species and amyloid fibrils to disfavor accumulation of oligomeric species and promote amyloid aggregation (*Figure 4E*). Without the N17 domain, ΔN becomes kinetically trapped in large and small oligomer states (*Figure 4B–D*, *Figure 4—figure supplement 1B*, *Figure 4E*). *Trans* addition of the N17 peptide can convert these trapped oligomers to fibrillar aggregates in vitro (*Figure 5B*) and accelerate mHtt-Ex1 aggregation in striatal-derived neurons (*Figure 5E–F*). N17 may act by increasing local mHtt concentration through inter-molecular interactions with the polyQ and N17 domains (*Atwal et al., 2011*; *Jayaraman et al., 2012a*; *Tam et al., 2009*), thereby nucleating amyloid formation (*Fiumara et al., 2010*; *Jayaraman et al., 2012b*) and promoting fibril bundling (*Figure 5B*). In addition N17 may induce a conformational switch in the polyQ tract, through direct intrachain interaction to induce its transition to β-sheet structures (*Kokona et al., 2014*; *Williamson et al., 2010*), that may be important for on-pathway aggregation of polyQ stretches (*Nagai et al., 2007*). The importance of the non-polar residues of N17 in promoting aggregation (*Figure 5E–F*) (*Tam et al., 2009*) could reflect the N17-N17 interaction interface, or a model where nonpolar N17 residues impact the conformation of the polyQ region.

In contrast to N17, the PRD destabilizes fibrils and stabilizes oligomeric species (*Figure 3*, *4*) and disfavors aggregation both in vitro and in vivo(*Figure 1*) (*Bhattacharyya et al., 2005*; *Tam et al., 2009*). Thus, removing the PRD in ΔP enhances the progression to amyloid fibrils and makes

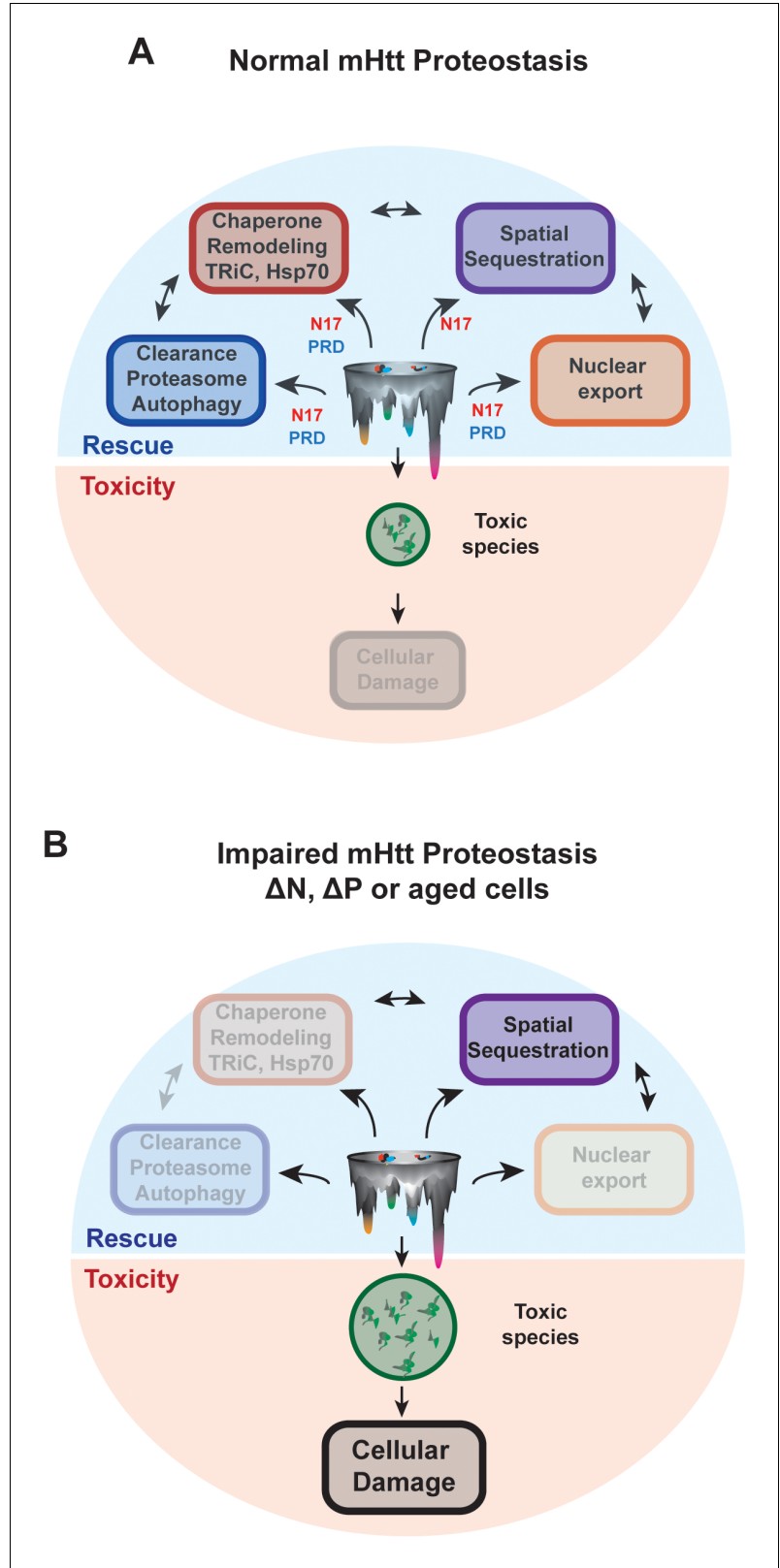

**Figure 9.** N17 and PRD control the conformational landscape and cellular fate of mHtt in vivo. (**A**) Normal HD proteostasis can occur through the coordination of several protein quality control pathways to clear toxic species from the cell, minimizing cellular damage: (1) protein degradation, (2) remodeling through molecular chaperones, (3) spatial sequestration through insoluble aggregation, and (4) export of toxic species from the nucleus. (**B**)

*Figure 9 continued on next page*

*Figure 9 continued*

Impaired HD proteostasis either in aged systems or altered mHtt translation products, creating ΔN or ΔP-like constructs. Here, normal proteostasis pathways – such as interaction through molecular chaperones, protein degradation machinery or integrity of proper nuclear-cytoplasmic trafficking – are diminished. The only mode of clearance of toxic species is through spatial sequestration, resulting in the accumulation of mHtt aggregates. However, the kinetics of this process may be insufficient to clear all toxic, soluble species, thereby allowing their accumulation and amplifying cellular damage.

oligomer states more transient (*Figure 4B–D*). ΔP amyloid fibrils are also more densely packed and more mechanically stable than amyloid fibrils containing a PRD region (*Figure 2A*, *3B–C*). Several models may explain these findings. The PRD could form a PolyProline-II (PPII) helix that propagates into the polyQ region to promote a more helical conformation in the polyQ rather than a β-sheet structure to disfavor aggregation (*Binette et al., 2016*; *Darnell et al., 2007*, *2009*). A dynamic PPII-helical PRD within the fibril could also interfere with fibril packing interactions (*Bugg et al., 2012*), perhaps by adopting a 'bottle brush' like architecture protruding from the amyloidogenic core of the fibril (*Isas et al., 2015*). Differences in the mechanical stabilities of fibrillar aggregates (*Figure 3*) may have implications for their frangibility in vivo and the generation of mHtt seeds competent for intercellular transmission and spread.

## Assessing the structure-proteotoxicity relationship in mutant Huntingtin

One perplexing aspect of CAG-repeat disorders like HD is why an expansion of the polyQ length beyond a certain threshold leads to neuronal toxicity (*Orr and Zoghbi, 2007*). One proposed theory is that toxicity arises from a feature in the repetitive ligand-binding units of a linear polyQ lattice (*Bennett et al., 2002*; *Owens et al., 2015*). In this sense, the longer the polyQ repeat, the more of these repetitive units may confer aberrant cellular interactions and cause pathogenesis. Alternatively, the 'emergent conformation' model proposes that a unique, toxic polyQ conformation appears at only pathogenic, expanded polyQ lengths (*Miller et al., 2011*).

Since mHtt constructs of identical polyQ length, but different flanking regions, have different toxicities (*Figure 6*), our results support the 'emergent conformation' model. The reduced toxicity of ΔNΔP implies the flanking regions contribute to generating an 'emergent' toxic conformation(s). Interestingly, all the toxic variants share a subset of conformers, as evident from their shared 3B5H10-reactivity. Even if the 3B5H10-reactive oligomer itself is not the toxic species, it appears diagnostic of a conformational pathway linked to toxicity.

Overall, these results have two major implications. Firstly, mHtt structural conformers and their cellular interactions are more relevant to pathogenesis than polyQ aggregation propensity. The Ex1, ΔN and ΔP variants are all highly toxic (*Figure 6C–E*) despite their different propensities to form insoluble aggregates or soluble oligomers both in vitro and in vivo (*Figure 1B-C*, *4B-E*, *7D*, *Figure 7—figure supplement 1B*). This suggests that only certain conformational sub-populations are neurotoxic (*Figure 7C–D*). Defining these toxic conformers is hindered by the limitations of current structural approaches, highlighting the need for improved methodologies to structurally define soluble mHtt states. Secondly, since polyQ alone generates toxic conformations inefficiently, the sequence context is key to the neurotoxic structural transition. Thus, the other polyQ-expansion disease causing proteins, such as ataxins (*Almeida et al., 2013*) must contain additional sequence elements that are functionally equivalent to N17 or the PRD. This finding highlights the need for understanding polyQ dynamics in the relevant protein context.

## The cellular fate of Huntingtin is synergistically determined by the polyQ flanking domains

Unexpectedly, we find that despite their distinct intrinsic effects in stabilizing either the oligomer or fibrillar states, N17 and PRD cooperate in vivo to determine mHtt fate (*Figure 9A*). mHtt may be prevented from exerting cellular damage through interaction with chaperones, clearance pathways as well as through sequestration into protective inclusion bodies, such as the aggresomes or IPOD (*Figure 9A*) (*Arrasate et al., 2004*; *Kaganovich et al., 2008*; *Kopito, 2000*). However, the

synergistic action of the N17 and PRD regions is essential to engage these proteostasis pathways that promote mHtt clearance and/or prevent formation of toxic species (*Figure 9*). While N17 contains the recognition site for CCT1 and is known to promote mHtt degradation, both N17 and PRD are required for efficient CCT1-mediated neuroprotection (*Figure 8B*, *Figure 8—figure supplement 1A*) and clearance of soluble mHtt species (*Figure 8D*). N17 is also essential to prevent the accumulation of mHtt in the nucleus (*Gu et al., 2015*), but deletion of PRD also causes nuclear-enrichment of mHtt toxic species, which may play an enhanced role in Htt pathogenesis (*Difiglia, 1997*).

We propose the observed synergy of the flanking regions in vivo is rooted in the conformational modulation exerted by the PRD to promote exposure of N17 (*Figure 8E*). This explains the paradoxical observations that, despite having an N17 domain, ΔP shows enhanced nuclear localization and poor neuroprotection from toxicity by TRiC/CCT1 (*Figure 7C*, *Figure 8A–B*). One model for this is the PRD may directly enhance N17 availability through transient interactions with each other (*Behrends et al., 2006*; *Caron et al., 2013*). Alternatively, the structural hindrance imposed by the PRD may prevent N17 from becoming buried in the early stages of oligomerization (*Figure 8D*, *Figure 8—figure supplement 1C*).

The key role of different proteostasis pathways in suppressing mHtt toxicity also suggests a model for the age-association of HD pathogenesis. Both proteostasis (*Brehme et al., 2014*; *Taylor and Dillin, 2011*) and nucleo-cytoplasmic trafficking (*D'Angelo et al., 2009*) decrease during aging. As a result, the N17-driven aggregation of mHtt into large, inert, amyloidogenic aggregates (*Figure 1B-F*, *5B–D*) remains the predominant available route for protection, causing neurons to become saturated with aggregates (*Waelter et al., 2001*). As a neuron's ability to clear mHtt declines, the ensuing accumulation of soluble species may exert further toxicity (*Figure 9B*). With age or the absence of N17, toxic oligomeric species accumulate, thereby exerting cellular damage. Of note, toxic, off-pathway oligomers are also reported for α-synuclein, emphasizing the potential toxicity of soluble species that cannot be sequestered into larger aggregates (*Chen et al., 2015*).

## Implication for Huntington's disease therapeutics

Our study suggests that effective therapeutics for HD should impact the conformers associated with toxicity. Better structural methods to study conformationally heterogeneous samples will be key to define these toxic species. The cellular proteostasis machinery, which can clearly identify and neutralize these toxic species, also offers therapeutic avenues. We show that TRiC, previously shown to interact with mHtt monomers (*Tam et al., 2009*), large oligomers (*Sontag et al., 2013*) and fibrillar aggregates (*Shahmoradian et al., 2013*), promotes clearance of small 3B5H10-reactive mHtt oligomers (*Figure 7E*) confirming its crucial protective role in HD. In addition, mHtt clearance rate in neurons is linked to their likelihood of survival (*Tsvetkov et al., 2013*). Thus, enhancing the ability of mHtt flanking regions to engage proteostasis pathways could provide highly specific and potent HD therapeutics.

One interesting question raised by our studies is whether ΔN and ΔP variants are generated during mHtt biosynthesis. Htt contains a methionine at residue 8 which could be used as an alternative translation initiation site, leading to a partial truncation of N17. Likewise, it has long been observed that proline-repeat stretches can stall translation (*Gutierrez et al., 2013*; *Ude et al., 2013*). For mHtt, this proline-dependent stalling may generate a partially truncated 'ΔP-like' mHtt-Ex1 variant. Future studies should test if these alternative translation mechanisms contribute to the generation of toxic mHtt fragments and HD pathogenesis.

## Materials and methods

### Protein expression and purification

pGEX-mHtt-Ex1-Q51 plasmid and mutants were constructed as previously described (*Tam et al., 2006*). Plasmids were expressed in Rosetta 2(DE3) pLysS competent cells (Agilent Technologies, Santa Clara, CA, USA) in LB media supplemented with carbenicilin and chloramphenicol. Cultures were induced with 1 mM IPTG for 2.5 hr at 16°C. For purification, pellets were resuspended in 50 mM sodium phosphate, pH 8.0; 150 mM NaCl; 1 mM EDTA and lysed using an Emulsiflex (Avestin, Ottawa, Canada). Lysate was incubated with GSH-Sepharose resin (GE Healthcare, Pittsburgh, PA, USA) and washed with 0.1% Triton, 500 mM NaCl, and 5 mM Mg-ATP before eluting protein

with 15 mM Glutathione. Protein was concentrated and buffer exchanged with 50 mM Tris-HCl, pH 8.0; 100 mM NaCl; 5% glycerol. Concentrated protein was 0.2 µm filtered before storage at −80℃.

### In vitro aggregation assays

Aggregation reactions were performed at concentration 3 µM of mHtt and 0.044 Units/µl acTEV (TEV) protease (Invitrogen, Carlsbad, CA, USA). Aggregation was conducted in TEV reaction buffer (Invitrogen) and incubated at 30℃. Time-points were taken accordingly.

### Filter trap assay

Time-points were combined in a 1:1 ratio with a 4% SDS, 100 mM DTT solution, boiled for 5 min at 95℃, and stored at −20℃. Samples were then filtered through a 0.22 µm cellulose acetate membrane (Whatman, Maidstone, United Kingdom) and washed with 0.1% SDS. Membrane was probed using an S-tag antibody (Abcam, Cambridge, United Kingdom).

### ThioflavinT assay

Aggregation reaction was prepared as above and combined with 12.5 µM ThioflavinT dye (Sigma-Aldrich, St. Louis, MO, USA). Reactions were transferred to a 3904 Corning plate and read with an Infinite M1000 plate reader (Tecan Systems, San Jose, CA, USA). Plate reader conditions were 30℃ incubation, 446 nm excitation, 490 nm emission, reading every 15 min.

### Finke-Watzsky (F-W) amyloid kinetics modeling

The F-W model predicts a sigmoidal function of the fractional concentration of a product as a function of time, $\theta(t)$. Such equation can be described by two independent parameters: (1) the time required to produce half of the fractional concentration ($t_{1/2}$), and (2) the rate of growth at that time ($v=\theta(t_{1/2})$). Therefore our F-W fitting function results:

$$\theta(t) = \frac{1}{1 + e^{-4v(t-t_{1/2})}} \tag{1}$$

Each curve was baseline corrected and normalized by its final value and then fitted to *Equation 1* by least-square fitting in Matlab.

### Live cell imaging

Live imaging was carried out on a Zeiss LSM 700 microscope (Carl Zeiss, Oberkochen, Germany) using epifluorescence. Cells were imaged 48 hr after transfection with 1 µg/ml Hoechst stain (Invitrogen) for 5 min prior to imaging. At least 150 cells were counted for each mHtt mutant. The percentage of cells with puncta was normalized with a number of transfected cells.

### Cryo-EM imaging and annotations

Time-points were flash frozen at $LN_2$ and kept at −80℃ for storage. Quantifoil 1.2/1.3 grids (Quantifoil 2/2 grids for ∆P fibers) (Electron Microscopy Sciences, Hatfield, PA, USA) were glow-discharged for 30 s. Each sample was thawed and deposited on EM grids. Grids were plunge-frozen using a Vitrobot Mark III (FEI, Hillsboro, Oregon, USA) with a '4 s, 1' blot setting in $LN_2$-cooled liquid ethane. Grids were stored at −80℃. Grids were loaded into JEOL JEM2010F (JEOL, Peabody, MA, USA) electron microscope. For each sample, approximately 15–25 images were collected on a Gatan 4k CCD camera at a defocus of −5 µm and 27,624x magnification with a sampling value of 5.43 Å per pixel. Representative 2D micrographs were converted from. DM3 to. PNG file type using Fiji software, then manually traced using a Wacom tablet on a separate invisible layer made atop each micrograph using Adobe Photoshop software. Detailed traces corresponding to zoomed-in regions were performed via Adobe Photoshop software by linear thresholding followed by selective manual removal of background using a Wacom tablet.

### Sonication assay

Aggregation reactions were prepared at 20 µM for Ex1, ∆P and ∆N∆P and 50 µM for ∆N. After 60 hr, aggregation reactions were spun at 16,000 ×g for 1 hr at room temperature to pellet fibers. The fiber pellet was resuspended in 1x TEV buffer (Invitrogen) and probe sonicated for various time

and power ratings with 1 s on/1 s off pulsing using a Fisher Scientific 120 W Sonic Dismembrator (Thermo Fisher Scientific, Waltham, MA, USA). The size of the sonicated fibers was measured by DLS using a Zetasizer Nano ZS (Malvern, Worcestershire, UK).

## Urea and formic acid denaturation assay

Aggregation reactions were prepared at 20 μM for Ex1 and ΔP. After 48h, aggregation reactions were spun at 16,000 ×g for 30 min at room temperature to pellet fibers. Supernatant sample was saved, and pellets were resuspended with 8M urea and incubated for 30 min at 37C, 1000 rpm shaking in a thermomixer. The samples were spun again with the same conditions. Supernatant sample was saved, and the pellets were resuspended with various concentrations of formic acid and incubated for 30 min at 37°C, 1000 rpm shaking in a thermomixer. Formic acid samples were dried by SpeedVac and resuspended in 8M urea. Samples were combined with 4x Laemlli buffer and 4M urea for running in SDS-PAGE gel and immunoblotting against the S-tag.

## Native and SDS AGE gels

Time-points were combined in a 1:1 ratio with sample buffer (150 mM Tris, pH 6.8, 33% glycerol, bromophenol blue, and 1.2% SDS only for the SDS-AGE samples) and stored at −80°C. Samples were then loaded on a 1% agarose gel with running buffer 25 mM Tris, 192 mM glycine, and 0.1% SDS for SDS-AGE gel. Gels were run at 125V until the dye front had migrated approximately 13 cm. The gel was then semi-dry blotted for 1 hr onto a PVDF membrane and probed with a S-tag antibody (Abcam), GFP antibody (Clontech, Mountain View, CA, USA), or 3B5H10 antibody (Sigma-Aldrich).

## N17 peptide transfection

Lyophilized N17 peptide (Genscript, Piscataway, NJ) was solubilized with PBS and sonicated for 30 min in an ice-bath before transfection. Peptide transfections were performed using the Protein Xfect transfection reagent (Clontech) according to the manufacturer's instructions.

## Mammalian cell culture and lysis

ST14a cells were cultured at 32°C, 5% $CO_2$ in DMEM media (Invitrogen) supplanted with 10% FBS and Pen/Strep. Cells were transfected using Lipofectamine 2000 (Invitrogen). Media was refreshed immediately before and within 24 hr post transfection. Cells were harvested 48 hr after transfection and lysed with 10 mM Tris, pH 7.5; 150 mM NaCl; 1 mM EDTA; 0.5% NP-40; 1 mM PMSF; 1x Protease Inhibitor (Roche, Basel, Switzerland). Protein concentration of lysate was measured by BCA assay (Thermo Fisher Scientific).

## Organotypic culture of brain slice explants

Hemi-coronal brain slices were prepared accordingly to established procedures (*Reinhart et al., 2011*). Briefly, postnatal day 10 Sprague–Dawley rat pups (Charles River, Wilmington, MA, USA) were used to prepare 250 μm thick hemi-coronal brain slices using vibratomes (Vibratome; Bannockburn, IL, USA). Animals were killed in accordance with NIH guidelines and under Duke IACUC approval and oversight. Slices were placed into 12-well plates on a semi-solid support consisting of culture medium (Neurobasal A medium supplemented with 15% heat-inactivated horse serum, 10 mM KCl, 10 mM HEPES, 100 U/mL penicillin/streptomycin, 1 mM sodium pyruvate, 1 mM L-glutamine, and 1 μM MK-801) set in 0.5% agarose. Slice cultures were maintained at 32°C in humidified incubators under 5% $CO_2$.

## Biolistic transfection and viability assay in brain slices

A modified Helios Gene Gun (Helios Gene Gun, Bio-Rad, Hercules, California) was used for particle-mediated gene transfer, or biolistics, of all mHtt plasmids into brain slices as described previously (*Lo et al., 1994*). Brain slices were co-transfected with a yellow fluorescent protein (YFP) expression plasmid in all cases, to image medium spiny neurons (MSNs). MSNs were identified by their location in the striatum and characteristic dendritic aborization. Brain slices were assessed for viability on day 3 or 4 after transfection by counting the number of healthy MSNs, as previously described (*Crittenden et al., 2010*). Briefly brain slices were imaged on a fluorescence

stereomicroscope (SteREO Lumar.V12, Carl Zeiss, Thornwood, NY) and qualified MSNs as 'healthy' if presenting a normal cell body diameter, a continuous expression of YFP within all cell compartments, and more than two discernable primary dendrites were scored as viable (*Crittenden et al., 2010*). Average numbers of healthy MSNs per brain slice explant (N=12 brain slices per condition) are shown as means ± SEM and statistical significance was assessed by an unpaired t test with Welch's correction using the GraphPad software.

### Immunostaining and confocal imaging

Transfected ST14a cells were immunostained with the 3B5H10 antibody. Briefly, cells were fixed with 4% PFA for 15 min at 4°C, then quenched with 10 mM $NH_4Cl$ for 5 min at room temperature. Cells were permeabilized with 0.2% Triton, then blocked for 1 hr with 5% Normal Donkey Serum (Jackson Immunoresearch, West Grove, PA). Cells were stained with 3B5H10 in 1:10,000 dilution for 2 hr, then Alexa Fluor 546 conjugate secondary antibody (Life Technologies, Carlsbad, CA) for 1 hr.

### Blue-Native PAGE gels

Time points were combined with 4x Native PAGE Sample Buffer (Invitrogen) and stored at −80°C. Samples were run on a 4–16% Bis-Tris gel (Invitrogen) at 150V for 150 min and transferred for 16 hr onto a PVDF membrane using the NuPAGE transfer buffer (Invitrogen).

### $^{35}$S radioactive pulse chase

ST14a cells were transfected with mHtt-Ex1-GFP constructs. 14 hr after transfection, cells were methionine-starved for 1 hr with 'methionine starvation media' (DMEM, high glucose, no methionine, dialyzed FBS, 1 mM L-glutamine, 1 mM L-cystine). Cells were then pulse labeled for 3 hr with methionine starvation media with 50 mCi/ml $^{35}$S-methionine (Perkin Elmer, Santa Clara, CA, USA). Then, cell media was replaced with rich DMEM high glucose media for the chase, which was in total 18 hr post transfection. Cells were harvested at respective time points and lysed with 50 mM Tris-HCl/7.5, 150 mM NaCl, 0.5% NP-40, 1x Roche Protease Inhibitor, 1 mM DTT, 1 mM PMSF. Lysate concentration was measured by BCA assay. mHtt protein was immunoprecipitated with a GFP nanobody. Protein mean lifetimes were calculated by normalizing radiogram signal intensity for each time point, then fitting the data to a nonlinear regression curve using the GraphPad software.

### Proteinase K and Trypsin digestion assays

Aggregation reactions were removed at the 3 hr time point and incubated with Proteinase K or Trypsin (Sigma-Aldrich) for 30 min on ice. Digestion was stopped with 1 mM PMSF and 1x Laemmli Sample Buffer. N17 antibody (Ab1) was kindly provided by Dr. Marian DiFiglia.

## Acknowledgements

We thank Drs. L Thompson and M DiFiglia for kindly providing cells and antibodies. We thank D Dunn and Drs. E Sontag, W Vonk, and T Kelly Rainbolt for experimental assistance and advice; Dr. P Dolan for help with figure preparation; and members of the Frydman lab for fruitful discussions. This work was supported by an NSF Graduate Fellowship and Gerhard Casper Stanford Graduate Fellowship to KS and NIH grants to JF, WC, and DCL.

## Additional information

### Funding

| Funder | Grant reference number | Author |
| --- | --- | --- |
| National Institute of General Medical Sciences | gm56433 | Koning Shen<br>Judith Frydman |
| National Institute of Neurological Disorders and Stroke | NS080514 | Barbara Calamini<br>Donald C Lo |
| NIH Office of the Director | pn2ey016525 | Sarah H Shahmoradian<br>Wah Chiu<br>Judith Frydman |

| National Institute of General Medical Sciences | gm103832 | Boxue Ma Sarah H Shahmoradian Wah Chiu |
| --- | --- | --- |
| Ellison Medical Foundation | | Jonathan A Fauerbach Judith Frydman |
| National Institute of Neurological Disorders and Stroke | NS092525 | Koning Shen Judith Frydman |

The funders had no role in study design, data collection and interpretation, or the decision to submit the work for publication.

**Author contributions**

KS, Conception and design, Acquisition of data, Analysis and interpretation of data, Drafting or revising the article; BC, JAF, Acquisition of data, Analysis and interpretation of data, Drafting or revising the article; BM, Acquisition of data, Drafting or revising the article; SHS, WC, DCL, Analysis and interpretation of data, Drafting or revising the article; ILSL, Helped do the experiments required during the revision process, Acquisition of data; JF, Conceived and directed project, Conception and design, Analysis and interpretation of data, Drafting or revising the article

**Author ORCIDs**

Koning Shen, http://orcid.org/0000-0003-2607-449X
Judith Frydman, http://orcid.org/0000-0003-2302-6943

**Ethics**

Animal experimentation: No human subjects. Animals were handled and killed in accordance with NIH guidelines and under approval and oversight of the Duke University institutional animal care and use committee (IACUC) to Don Lo. Protocol number A147-14-06.

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
