## [Decision Letter]

Thank you for submitting your article "The polyQ flanking regions control the conformational landscape, proteostasis, and neurotoxicity of mutant Huntingtin" for consideration by *eLife*. Your article has been reviewed by two peer reviewers, one of whom Jeffery W Kelly (Reviewer#1), is a member of our Board of Reviewing Editor, and the evaluation has been overseen by John Kuriyan as the Senior Editor.

The reviewers have discussed the reviews with one another and the Reviewing Editor has drafted this decision to help you prepare a revised submission.

1) Protein aggregation seems to be the driver of numerous neurodegenerative diseases, including Alzheimer's, Parkinson's and Huntington's disease. How the aggregates cause loss of post-mitotic tissue is not clear. This paper seeks a structure-proteotoxicty assessment.

2) Koning Shen and colleagues use domain truncations of mutant huntingtin exon 1 to generate aggregates with different properties. Consistent with previous reports, they find the flanking regions of the extended polyQ tract strongly influence the properties of the aggregates formed. Combining various approaches, including elegant electron microscopy experiments with an antibody that is thought to recognize a toxic huntingtin species (Miller et al. Nat Chem Biol 2011), they do a great job of characterizing the fibrils resulting from each domain deletion in detail. The in vitro results presented are an important extension of published work in this field, and are well worth publishing. We are more reserved about the in vivo sections of this manuscript. There is no clear overall theme, and it is not clear that the experiments presented add much in terms of impact, as the manuscript is organized in a way to suggest that the characteristics of the fibrils are not (or only weakly) correlated with toxicity. We would therefore favor a version of the manuscript that focuses mainly on the characterization of the different aggregate species in vitro.

Strengths of the paper:

The authors present multiple lines of evidence suggesting that deleting the PRD domain (ΔP) forms amyloid fibrils that are SDS insoluble in vitro and in striatal neurons. ΔP adopts a bundled fibril densely packed amyloid structure by EM, an ensemble of aggregate stratures that largely lack oligomers based on native gels.

Multiple lines of evidence suggest that deleting the N17 domain (ΔN) forms predominantly oligomers that are non-amyloid, but fibrillar, based on low ThT binding, their EM morphology and the smear produced in Native-AGE and 0.1% SDS-AGE.

The sonication approach to compare aggregate stability produced by Q-51 expanded exon 1, ΔP and ΔN is clever, it might be interesting to compare urea denaturation susceptibility.

The EM work is beautiful, however a statistically significant approach to quantify the fibrils vs. oligomers vs additional aggregate structures would impart confidence to the reader with regard to the aggregate strtural distributions that are being made in the central brain slice experiments.

Essential revisions:

We would favor a version of the manuscript that focuses mainly on the characterization of the different aggregate species in vitro.

The comparisons of the toxicity of full-length Q-51 expanded exon 1, ΔP and ΔN were not different enough to convince either reviewer that the oligomers were the main toxic species. If the authors feel that the in vivo work should be part of the revised paper; perhaps the authors can attempt to enrich the oligomers by gel filtration chromatography with variable degrees of light crosslinking to see whether it is truly the oligomers that are driving toxicity in cell culture model(s) and possibly in brain slices, if the oligomers are taken up? I realize that some aggregate remodeling may take place.

The other experiment that would have increased my support for the oligomer toxicity hypothesis would be to characterize the aggregate structures bound to antibody 3B5H10 by EM. An oligomer probe/antibody that blocked toxicity in the brain slice assays would be very powerful.

Please consider as many of the details addressed below in the reviews that agree in large part when revising this manuscript.

Reviewer #1:

*eLifeeLife*In the manuscript entitled "The polyQ flanking regions control the conformational landscape, proteostasis, and neurotoxicity of mutant Huntingtin" the Frydman group explore structure-proteotoxicity relationships in neurons and brain slices, studies which are central to understand how and why the process of aggregation causes the loss of neurons in Huntington's disease. The Frydman group studies 4 variants of a Q-51 expanded exon 1, one lacking the N17 amyloid promoting domain, a Q-51 expanded exon 1 lacking the PRD amyloid inhibiting domain, a Q-51 expanded exon 1 lacking both domains flanking the Q-51 expanded exon 1, as well as full-length Q-51 expanded exon 1.

The authors present multiple lines of evidence suggesting that deleting the PRD domain (ΔP) forms amyloid fibrils that are SDS insoluble in vitro and in striatal neurons. ΔP adopts a bundled fibril densely packed amyloid structure by EM, an ensemble of aggregate stratures that largely lack oligomers based on native gels.

Multiple lines of evidence suggest that deleting the N17 domain (ΔN) forms predominantly oligomers that are non-amyloid, but fibrillar, based on low ThT binding, their EM morphology and the smear produced in Native-AGE and 0.1% SDS-AGE.

The sonication approach to compare aggregate stability produced by Q-51 expanded exon 1, ΔP and ΔN is clever, it might be interesting to compare urea denaturation susceptibility.

The EM work is beautiful, however a statistically significant approach to quantify the fibrils vs. oligomers vs additional aggregate structures would impart confidence to the reader with regard to the aggregate strtural distributions that are being made in the central brain slice experiments.

What was disappointing to me, and I am sure the authors, is the comparisons of the toxicity of full-length Q-51 expanded exon 1, ΔP and ΔN. That the toxicities were very similar does not convince this reviewer that the authors can conclude that oligomers are the principal toxic species in Huntington's disease.

Why did the authors not try to enrich the oligomers by gel filtration chromatography with variable degrees of light crosslinking to see whether it is truly the oligomers that are really driving toxicity in cell cultures and possibly in brain slices if the oligomers are taken up? I realize that some aggregate remodeling may take place.

The other experiment that would have increased my support for the oligomer toxicity hypothesis would be to characterize the aggregate structures bound to antibody 3B5H10 by EM. Having an oligomer probe that blocked toxicity in the brain slice assays would be very powerful.

I am on the fence whether this is an *eLife* paper. While the results reported required an almost heroic amount of work, one is left unconvinced that the oligomers are the toxic species, which is but one of many conclusions of the paper that could benefit from an increase in focus.

Reviewer #2:

Formation of protein aggregates is the common denominator of numerous neurodegenerative diseases, including Alzheimer's, Parkinson's and Huntington's disease. How the aggregates cause disease is not clear, though, and the biophysical properties that characterize the most toxic aggregate species remain to be defined. Answers to these questions will be relevant in the design of strategies to combat these diseases.

Koning Shen and colleagues use a set of truncations of mutant huntingtin exon 1 to generate aggregates with different properties. Consistent with previous reports they find the flanking regions of the extended polyQ tract to strongly influence the properties of the aggregates formed. Combining various techniques, including elegant experiments with an antibody that is thought to recognize a toxic huntingtin species (Miller et al. Nat Chem Biol 2011), they do a great job of characterizing the resulting fibrils in detail. The in vitro results presented are an important extension of published work in this field, and are well worth publishing.

I am a bit more reserved about the in vivo sections of this manuscript. There is no clear overall theme, and I don't believe that the experiments presented add much in terms of impact, as the manuscript is organized in a way to suggest that the characteristics of the fibrils are not (or only weakly) correlated with toxicity. I would therefore favor a version of the manuscript that focuses mainly on the characterization of the different aggregate species in vitro.

---

## [Author Response]

*The reviewers have discussed the reviews with one another and the Reviewing Editor has drafted this decision to help you prepare a revised submission.*

*1) Protein aggregation seems to be the driver of numerous neurodegenerative diseases, including Alzheimer's, Parkinson's and Huntington's disease. How the aggregates cause loss of post-mitotic tissue is not clear. This paper seeks a structure-proteotoxicty assessment.*

*2) Koning Shen and colleagues use domain truncations of mutant huntingtin exon 1 to generate aggregates with different properties. Consistent with previous reports, they find the flanking regions of the extended polyQ tract strongly influence the properties of the aggregates formed. Combining various approaches, including elegant electron microscopy experiments with an antibody that is thought to recognize a toxic huntingtin species (Miller et al. Nat Chem Biol 2011), they do a great job of characterizing the fibrils resulting from each domain deletion in detail. The* in vitro *results presented are an important extension of published work in this field, and are well worth publishing. We are more reserved about the* in vivo *sections of this manuscript. There is no clear overall theme, and it is not clear that the experiments presented add much in terms of impact, as the manuscript is organized in a way to suggest that the characteristics of the fibrils are not (or only weakly) correlated with toxicity. We would therefore favor a version of the manuscript that focuses mainly on the characterization of the different aggregate species* in vitro.

We thank the Reviewers and Editor for their thoughtful comments on our manuscript. The Reviewers were overall positive about the timeliness and relevance of our study, stating that: “Combining various approaches, including elegant electron microscopy experiments with an antibody that is thought to recognize a toxic huntingtin species (Miller et al. Nat Chem Biol 2011), they do a great job of characterizing the fibrils resulting from each domain deletion in detail. The in vitro results presented are an important extension of published work in this field, and are well worth publishing.”

The Reviewers were more reserved about our in vivo analyses, stating that: “There is no clear overall theme” and also found that the “manuscript is organized in a way to suggest that the characteristics of the fibrils are not (or only weakly) correlated with toxicity.” In retrospect, we agree with the reviewers that the data, as originally organized, did not cleanly tie the biophysical and cell biological characterization of the mutant Huntingtin (mHtt) conformational landscape. We have made an effort to re-organize and refocus our MS to more clearly explain our overall view, which was to both characterize the conformational landscape of Htt, and to link the biophysical and structural properties of Htt to its proteostasis in vivo and its toxicity to neurons. As part of this revision, we have also changed the title to “Control of the structural landscape and neuronal proteotoxicity of mutant Huntingtin by domains flanking the polyQ tract”, which we feel embodies better the focus of our work.

One major goal of our work is to bridge the disconnect between the biophysical nature of aggregation-prone proteins and their toxicities. Many studies focus on the biophysical properties of aggregation-prone proteins, while others focus on their toxicities and behavior in cell culture, leaving a large gap in understanding how the biophysical nature of these aggregating proteins leads to their cellular consequences. Here we address the challenge of integrating these biochemical and biophysical approaches and link them to toxicity in a relevant brain model. However, the complexity of moving back and forth between systems was evident from the disjointed nature of our original manuscript. Our current revision focuses on the novel in vitro structural insights, while simplifying the integration of our in vivo analysis.

We have also softened our discussion of the nature of the toxic species. Our study shows that toxicity may not derive from just fibrils or oligomers but likely in specific conformations (or conformational ensembles) as shown by the selective reactivity of toxic variants to the 3B5H10 antibody. While we feel that the 3B5H10 reactive species illustrates this point, we do not want to state that this is “the” toxic conformation, since there may be toxic species that we cannot detect due to a lack of conformational probes. This may include specific conformations of fibrils, since we show here that there is much conformational diversity among fibrils. We have now expanded on this discussion in the manuscript..

What we also show in the final in vivo figures of the manuscript is that the cellular chaperone and clearance machineries can interact with 3B5H10-reactive species in suppressing their toxicity. A novel insight of these experiments is that the flanking regions are not only essential in this recognition but also act synergistically to engage the cellular machinery, rooted in the intrinsic conformational interplay between N17 and the PRD. We hope that we have succeeded in the revision to capture the role the polyQ flanking regions play in controlling complex interplay between the biophysics and cell biology leading to the cellular fate mHtt and that these results will be of great interest to those studying other neurodegenerative diseases.

*Strengths of the paper:*

*The authors present multiple lines of evidence suggesting that deleting the PRD domain (ΔP) forms amyloid fibrils that are SDS insoluble in vitro and in striatal neurons. ΔP adopts a bundled fibril densely packed amyloid structure by EM, an ensemble of aggregate stratures that largely lack oligomers based on native gels.*

*Multiple lines of evidence suggest that deleting the N17 domain (ΔN) forms predominantly oligomers that are non-amyloid, but fibrillar, based on low ThT binding, their EM morphology and the smear produced in Native-AGE and 0.1% SDS-AGE.*

*The sonication approach to compare aggregate stability produced by Q-51 expanded exon 1, ΔP and ΔN is clever, it might be interesting to compare urea denaturation susceptibility.*

We thank the reviewers for their comment. Based on their suggestion, we have compared the sensitivity of the Ex1 and ∆P fibrils to urea and formic acid denaturation (Figure 3—figure supplement 2). We found the results to nicely highlight important aspects for biophysical characteristics of mHtt fibrils.

First, we found that both Ex1 and ∆P mHtt-Ex1 fibrils were very resistant to urea denaturation. Despite their differential sensitivity to mechanical disruption, isolated Ex1 and ∆P mHtt-Ex1 fibrils were both completely refractory to 8M urea treatments (shown in Figure 3—figure supplement 2 is 8M urea treatment for 30 min at 37 °C but other conditions were also tested).

We next examined the differential sensitivity of these fibrils to formic acid treatments of increasing severity (Figure 3—figure supplement 2). Only 100% formic acid was able to significantly solubilize the mHtt fibrils formed by either the Ex1 or ∆P variants.

These new experiments show that, despite their very distinct mechanical stabilities, mHtt aggregates of either Ex1 or ∆P are so stable to chemical denaturation that they require 100% formic acid for significant – though only partial – solubilization.

*The EM work is beautiful, however a statistically significant approach to quantify the fibrils vs. oligomers vs additional aggregate structures would impart confidence to the reader with regard to the aggregate strtural distributions that are being made in the central brain slice experiments.*

This is an excellent suggestion and we now include a quantification of the structural features of the fibrils in Figure 2—figure supplement 2. These new analyses nicely bolster our conclusion that the flanking regions contribute to the distinct structure and architecture of these fibrils. By measuring the length and width of the fibrils, we quantitatively support our model of how the N17 and PRD domains contribute to mHtt fibrillar structure.

With regards to the distribution between fibrillars and other soluble species, we have now included a biochemical analysis (Figure 3—figure supplement 2). After 48 hr of aggregation, all of the mHtt Ex1 or ∆P are quantitatively depleted from the soluble fraction (note complete absence of mHtt in the initial Sup).

However, with regards to EM analyses of oligomeric species, unfortunately, current cryoEM capabilities don’t have the ability to characterize (let alone quantify) these mHtt oligomers. We have tried extensively to use cryoEM to visualize oligomers, with and without antibody labeling, but their much smaller size and vast structural heterogeneity preclude their reliable identification on the grids. Accomplishing this is the subject of an on-going collaboration project with Dr. Wah Chiu’s lab at Baylor College of Medicine. We hope to report these results in the near future, but this was out of scope for the current study. We now comment on the urgent need of novel structural approaches to understand the structures of these soluble species.

*Essential revisions:*

*We would favor a version of the manuscript that focuses mainly on the characterization of the different aggregate species* in vitro.

We have restructured the manuscript to focus more on the biophysical impact of the flanking regions on the Htt aggregation landscape. In addition to incorporating the experiments suggested by the Reviewers, we have also restructured the in vivo data to better integrate it with the biophysical analyses of the different mHtt variants.

*The comparisons of the toxicity of full-length Q-51 expanded exon 1, ΔP and ΔN were not different enough to convince either reviewer that the oligomers were the main toxic species.*

We agree that this experiment was not well presented in the original version of the manuscript, and we have attempted to clarify both its importance and our conclusions.

We feel it is important to include the toxicity experiment, because it is the first time that the toxicities of these various flanking region variants are compared side-by-side in a relevant neuronal model system, rather than isolated cells in culture.

In addition, these results are important, given the many conflicting models of “aggregate” or “oligomer-only” toxicity. We find that the mHtt variant that forms very few fibrils in vitro is as toxic as the one that forms few oligomers. Even more surprising, a polyQ tract of identical length has almost no discernible toxicity. We feel these results dispel the more simplistic models of toxicity, which we feel is an important conclusion in moving the field forward.

As to the identity of the toxic species, the Reviewers are correct that we cannot state that a 3B5H10 reactive oligomer is the toxic species. In principle, given the heterogeneity of mHtt aggregate species, it is also possible that a certain subset of fibrils is also toxic. What we can suggest based on our data, is that a subset of conformations must be toxic. At the moment, we unable identify them beyond the very few available conformation-sensitive antibodies. We expand more on the relevance of these 3B5H10-reactive oligomer species in Figure 10 and in this Response letter.

*If the authors feel that the* in vivo *work should be part of the revised paper; perhaps the authors can attempt to enrich the oligomers by gel filtration chromatography with variable degrees of light crosslinking to see whether it is truly the oligomers that are driving toxicity in cell culture model(s) and possibly in brain slices, if the oligomers are taken up? I realize that some aggregate remodeling may take place.*

We did try to develop a system to directly test whether oligomers or fibrils added directly to cells in culture or striatal brain slices can mediate toxicity. This would open the door to the identification of toxic species as well as therapeutics that block toxicity. As described below, our initial attempts at these experiments are promising, and do indeed support the toxic nature of 3B5H10 oligomers, but the system will require extensive optimization (see Figure 10). Given that our effects – while statistically significant – are still small, we have decided against including these results in our manuscript. We prefer to wait for the system to become more robust before presenting it to the public.

We used StHdh Q7/111 striatal neurons and measured cell toxicity by the LDH assay (Figure 10). We generated oligomers from the supernatant of a 3h aggregation reaction; fibrils were obtained from a 24h aggregation reaction. Unfortunately, oligomers cannot survive a gel filtration step, complicating further purification steps. Oligomers and fibrils were added to striatal neurons using the Xfect protein transfection kit (Clontech) and toxicity was assessed 48 hours after protein transfection.

Overall, we observed statistically significant but small effects: oligomers were significantly toxic to cells, whereas the fibrils were not (Figure 10). Just the soluble oligomer fraction (30 min, 16000 x g) was also significantly toxic when added to these striatal neurons (Figure 10). Notably, pre-incubating these soluble oligomers with the 3B5H10 conformational antibody abrogated their toxicity, suggesting that the 3B5H10 antibody indeed blocks a toxic determinant in these soluble species (Figure 10). Overall, we believe that these data impart confidence to our conclusion that a specific sub-population of soluble species is toxic, and this may include a population of oligomers that have a 3B5H10-reactive conformation.

Unfortunately, the effects, while statistically significant are still very small (Figure 10). We believe the problem lies in the inefficient uptake of exogenously added protein. Indeed, a test using Cy3-labeled mHtt-Ex1 (Figure 10) followed by confocal imaging of neurons showed that very little of Cy3-protein enters the cells (Figure 10). The finding that fewer than 10% of the cells uptake mHtt could explain the small effects observed in the toxicity assay. We believe that this promising assay is one of the first experiments demonstrating increased toxicity in striatal neurons upon mHtt-Ex1 addition. However, optimization of this assay is beyond the timeframe and scope of this study.

Given the above results, we were concerned that the uptake of the 3B5H10 antibody into these brain slices would be very low and thus these experiments were deemed very unlikely to yield interpretable results.

With regards to crosslinking, the Reviewer is correct that aggregate remodeling will take place. Because Huntingtin aggregates very quickly compared to other amyloidogenic proteins, even light crosslinking (0.05% glutaraldehyde) propels oligomeric species into aggregates rather than trapping them. This is an experiment we have tried many times and have only been able to successfully crosslink Htt oligomers with a short polyQ length (17 glutamine residues).

Finally, we agree with the Reviewer that separation of different sizes and populations of mHtt soluble species is a worthwhile effort, especially to characterize populations of soluble mHtt that may differentially contribute to toxicity. However, we have had little success eluting mHtt from gel filtration columns. We have tried a multitude of resins, with only limited success. While this endeavor would take beyond the timeframe given to revise this paper, we agree that this would be an important step forward for the field.

Author response image 1.Exogenous addition of mHtt-Ex1 oligomers and aggregates to striatal cells**A**. **i**. Experimental schematic of generating mHtt-Ex1 oligomers and aggregates in vitro and their exogenous addition to StHdhQ7/111 knock-in striatal cells. mHtt-Ex1 species were added to cells using protein transfection (Xfect kit, Clontech). Toxicity was assessed via LDH assay (Promega) 48h after their addition. **ii**. Measurement of LDH release as a proxy for cell death. Buffer control was the original sample buffer for the mHtt-Ex1 in vitro species. Data are mean ± SD. *p < 0.05. **B**. **i**.Experimental schematic. Experimental set-up was similar to (**A**) with the addition of a spin to procure just the soluble supernatant of the aggregation reaction after 3h. The supernatant was also pre-incubated with the 3B5H10 polyQ conformational antibody for 1h at RT to allow binding between any soluble oligomeric species and the 3B5H10 antibody. **ii**. Measurement of LDH release as a proxy for cell death. Data are mean ± SD. *p < 0.05, **p < 0.01 **C**.Left: Experimental schematic. mHtt-Ex1 was labeled with fluorescent Cy3-maleimide dye, then added to StHdh- Q7/111 striatal cells using the Xfect protein transfection kit, similar to (**A**) and (**B**). After 12h, cells were fixed and imaged via confocal microscopy (Right) to determine proportion of mHtt-Ex1 that gets into cells.**DOI:**
http://dx.doi.org/10.7554/eLife.18065.022

*The other experiment that would have increased my support for the oligomer toxicity hypothesis would be to characterize the aggregate structures bound to antibody 3B5H10 by EM. An oligomer probe/antibody that blocked toxicity in the brain slice assays would be very powerful.*

As described above, visualizing 3B5H10-bound oligomers by EM would be technically very challenging given their low molecular weight (100-200 kDa), their shape heterogeneity and small size. With this approach, the antibody would be the main species observed. An alternative experiment along this line of thought would be to immunoprecipite oligomeric species from an aggregation reaction using the 3B5H10 antibody, similar to as was done in Peters-Libeu et al., JMB 2012 from the Finkbeiner group at UCSF. Immunoprecipitation from either in vitro aggregation reactions or cellular lysates would allow for structural or biochemical analysis of the 3B5H10 reactive species. However, accomplishing this experiment would be a structural project in its own right and out of the scope for the current manuscript.

Furthermore, while our other experiments lend confidence that the 3B5H10-reactive oligomers are relevant to toxicity (Figure 10), we have restructured the manuscript to de-emphasize the identification of a specific toxic species in favor of a more general conclusion that toxicity may stem from only specific sub-populations of conformations. Our revised manuscript highlights the complexity of different Huntingtin conformations – some of which may be toxic and some of which may not – and merely suggests a particular population such as the 3B5H10-reactive small oligomers may be toxic. Because the oligomers are such a heterogeneous population, we do not want to imply that all oligomers are toxic and fibrils are not. There may be other conformations of mHtt, both fibrillar and oligomeric, that contribute to toxicity but go undetected in our study due to lack of proper detection methods. Importantly, our data highlight the need to develop structural tools that can characterize different sub-populations of mHtt species in order to identify more populations of toxic mHtt conformations.